# Universal Off-Policy Selection for Human-Centric Systems via Participant Sub-grouping

## Abstract

Human-centric tasks like healthcare and education are characterized by *heterogeneity* among patients and students, resulting in different disease trajectories and learning styles that require personalized treatments or instructional interventions for specific subgroups. When deploying reinforcement learning (RL) for such tasks, off-policy selection (OPS) is essential, since it closes the loop by selecting and evaluating RL-induced policies offline, without the need for any online interaction with the participants. Many pre-existing OPS methods, however, do not consider the heterogeneity among the participants. In this work, we introduce a universal off-policy selection (UOPS) approach to address the issue of participant heterogeneity by taking a multi-step approach. Initially, it divides the participants into sub-groups, grouping together those who exhibit similar behaviors. Subsequently, it acquires OPS criteria tailored to each of these sub-groups. Consequently, when new participants come, they will receive policy recommendations based on the sub-groups they align with. This methodology enhances the adaptability and personalization of the RL system, ensuring that policy selections align more closely with the unique characteristics of each participant or group of participants. We evaluate UOPS' effectiveness through two applications: an intelligent tutor system that has been used in classrooms for over eight years, as well as a healthcare application for sepsis treatment and intervention. In both applications, UOPS shows significant improvements in students' learning and patient outcomes.

## 1 Introduction

Human-centric systems (HCSs), *e.g.*, used in healthcare facilities (Raghu et al., 2017; Namkoong et al., 2020; Gao et al., 2020) and intelligent education (IE) (Chi et al., 2011; Koedinger et al., 1997; VanLehn, 2006), have widely employed reinforcement learning (RL) to enhance user experience by improving outcomes of disease treatment, knowledge gaining, etc. Specifically, RL has been used in healthcare to automate treatment procedures (Raghu et al., 2017), or in IE that can induce policies automatically adapting difficulties of course materials and helping students to setup and refine study plans to improve learning outcomes (Liu et al., 2022; Zhou et al., 2022). Though various existing offline RL methods can be adopted (Haarnoja et al., 2018; Kumar et al., 2020; Chen et al., 2021) for policy optimization, validation of policies' performance is often conducted by online testing (Silver et al., 2018; Wurman et al., 2022; Vinyals et al., 2019; Fu et al., 2021b). Given the long testing horizon (*e.g.*, several years, or semesters, in healthcare, and IE, respectively) and the high cost of recruiting participants, online testing is considered exceedingly time- and resource-consuming, and sometimes could even be hindered by protocols overseeing human involved experiments, *e.g.*, performance and safety justifications need to be provided before new medical device controllers can be tested on patients (Parvinian et al., 2018).

Recently, off-policy evaluation (OPE) methods have been proposed to tackle such challenges by estimating the performance of target (evaluation) RL policies with offline data, which only requires the trajectories collected over behavioral polices given *a priori*; similarly, off-policy selection (OPS) targets to determine the most promising policies, out of the ones trained with different algorithms or hyper-parameter sets, that can be used for online deployment Chandak et al. (2022); Fu et al. (2021a); Nie et al. (2022); Yang et al. (2022); Zhang & Jiang (2021). However, most existing OPS and OPE methods are designed in the context of homogenic agents, such as in robotics or games, where characteristics of the agents can be captured by their specifications, which are in general assumed

fully known (*e.g.*, degree of freedom, angular constraint of each joint). In contrast, in HCSs, the participants can have highly diverse backgrounds, where each person may be associated with unique underlying characteristics that are not straightforward to be captured individually; due to the partial observability of participants' mind states and the limited size of the cohort that can be recruited for experiments with HCSs. For example, patients participated in healthcare research studies could have different health/disease records, while the students using an intelligent tutoring system in IE may have different mindsets toward studying the course. As a result, the optimal criteria for selecting the policy to be deployed to each participant can vary, and, more importantly, it would be intractable for existing OPS/OPE frameworks to determine what the policy selection criteria would be for a new participant who just joined the cohort.

In this work, we introduce universal off-policy selection (UOPS), which addresses the problem of determining the OPS criteria needed for each new participant joining the cohort (*i.e.*, at $t = 0$ only, or without using information obtained from $t >= 1$ onwards), assuming that we have access to offline trajectories for a small batch of participants *a priori*, *i.e.*, the offline data. Specifically, it first partitions the participants from the offline dataset into sub-groups, clustering together the ones pertaining to similar behaviors. Then, an unbiased value function estimator, with bounded variance, is developed to determine the policy selection criteria for each sub-group. At last, when new participants join, they will be recommended with policies selected according to the sub-groups they fall within. Note that UOPS is distinguished from typical off-policy selection (OPS) setup in the sense that, the major goal of prior OPS approaches is to select the best policy over the entire population, while UOPS aims to decide the best policy for each student who arrives to the HCS on-the-fly, leveraging the information observed at the initial step ($t = 0$) only.

The key contributions of this work are summarized as follows: (*i*) We introduce the UOPS framework which is critical for closing the gap between offline RL policy optimization and OPS in selecting policies to be deployed over individual participants in HCSs. To the best of our knowledge, this is the first framework that considers the new participant arrival's problem in the context of OPS. (*ii*) We conduct extensive experiments to evaluate UOPS in a ***real-world IE system***, with 1,288 students participating over 5 years. Results have shown that, with the help of UOPS, it improved the learning outcomes by $208\%$ compared to policy selection criteria hand-crafted by instructors. Moreover, it leads to $136\%$ increased outcome compared to policies selected by existing OPS methods. (*iii*) UOPS is also evaluated against an important healthcare application, *i.e.*, septic shock treatment (Raghu et al., 2017; Namkoong et al., 2020; Oberst & Sontag, 2019), where it can accurately identifying the best treatment policies to be deployed to incoming patients, and outperforms existing OPS methods.

## 2 Universal Off-Policy Selection (UOPS)

In this section, we introduce the UOPS method, which determines the policy to be deployed to new participants that join an existing cohort, conditioned only on their initial states. Specifically, the participants pertaining to the offline dataset are partitioned into sub-groups according to their past behavior. Then, a variational auto-encoding (VAE) model is used to generate synthetic trajectories for each sub-group, augmenting the dataset and improving the state-action coverage. Moreover, an unbiased value function estimator, with bounded variance, is developed to determine the policy selection criteria for each sub-group. At last, when new participants join, they will be recommended with the policies conditioned on the sub-groups they fall within respectively. We start with a sub-section that introduces the problem formulation formally.

### 2.1 Problem Formulation

The HCS environment is formulated as a human-centric Markov decision process (HI-MDP), which is a 7-tuple $(\mathcal{S}, \mathcal{A}, \mathcal{P}, \mathcal{S}_0, R, \mathcal{I}, \gamma)$. Specifically, $\mathcal{S}$ is the state space, $\mathcal{A}$ is the action space, $\mathcal{P} : \mathcal{S} \times \mathcal{A} \rightarrow \mathcal{S}$ defines transition dynamics from the current state and action to the next state, $\mathcal{S}_0$ defines the initial state distribution, $R : \mathcal{S} \times \mathcal{A} \rightarrow \mathbb{R}$ is the reward function, $\mathcal{I}$ is the set of participants involved in the HCS, $\gamma \in (0, 1]$ is discount factor. Episodes are of finite horizon $T$. At each time-step $t$ in *online* policy deployment, the agent observes the state $s_t \in \mathcal{S}$ of the environment, then chooses an action $a_t \in \mathcal{A}$ following the *target (evaluation)* policy $\pi$. The environment accordingly provides a reward $r_t = R(s_t, a_t)$, and the agent observes the next state $s_{t+1}$ determined by $\mathcal{P}$. A *trajectory* is denoted as $\tau_\pi^{(i)} = [\ldots, (s_t^{(i)}, a_t^{(i)}, r_t^{(i)}, s_{t+1}^{(i)}), \ldots]_{t=1}^T$. Moreover, we consider having access to a historical trajectory set (*i.e.*, offline dataset) collected under a *behavioral* policy $\beta \neq \pi$, $\mathcal{D}_\beta = \{..., \tau_\beta^{(i)}, ...\}_{i=1}^N$,

which consist of $N$ trajectories. We first make two assumptions in regards to the correspondence between trajectories and participants, and the initial state distribution for each participant, respectively.

**Assumption 1** (Trajectory-Participant Correspondence). *As a participant in human-centric experiments is in general unlikely to undergo exactly the same procedure more than once under the topic being studied, we assume that there exist a unique correspondence between each trajectory $\tau^{(i)}$ and the participant ($i \in \mathcal{I}$) from which the trajectory is logged.*

We henceforth can use $i$ to refer to index either a trajectory from the offline dataset, or the corresponding participant, depending on the context.

**Assumption 2** (Independent Initial State Distributions). *The initial state of each trajectory $s_0^{(i)} \in \tau^{(i)}$, corresponding to a unique (the i-th) participant following from the assumption above, is sampled from an initial state distribution $\mathcal{S}_0(i)$ conditioned on i-th participant's characteristics and past records (i.e., specific to the i-th trajectory), and is independent from all other $\mathcal{S}_0(j)$'s where $j \in [1, N]\backslash i$.[1]*

The assumptions above reflect the scenarios that are specific to HCS – for example, a patient is unlikely to be prescribed the same surgery twice. Even if the patient has to undergo a follow-up surgery that is of the similar type (*e.g.*, mostly seen in trauma or orthopedics departments), the second time when the patient comes in he/she will start with a rather different initial state, since the pathology may have already been intervened as a result of the last visit. Consequently, one can treat such a visit as a new (synthetic) participant who just join and has the health record same as the one updated after the last visit. In other words, a *participant* being considered in this paper can be generalized, *e.g.*, to a *hospital visit*, or a *student* participating in a *specific course* supported by intelligent education (IE) systems, depending on the context. Moreover, assumption 2 directly follows from the philosophy illustrated in assumption 1 – the initial state of each trajectory depend on the corresponding participant's unique characteristics and historical records before joining the experiment/cohort, and can be considered mutually independent across all participants. Now we define the goal for UOPS.

**Problem 1.** *The goal of UOPS is to select the best policy $\pi$ from a set of **pre-trained** (candidate) policies $\boldsymbol{\Pi}$, $\pi \in \boldsymbol{\Pi}$, for each of the new participants $i' \in \{N+1, N+2, \dots\}$ joining (i.e., arriving at) the HCS with an observable initial state $s_0 \sim \mathcal{S}_0(i')$ (but the rest of the trajectory remain unobservable), that maximizes the expected accumulated return $V^\pi$, $\max_{\pi \in \boldsymbol{\Pi}} V^\pi$, over the full horizon $T$; here $V^\pi = \mathbb{E}_{s_0 \sim \mathcal{S}_0(i'), (s_{t>0}, a_{t>0}) \sim \rho^\pi, r \sim R}[\sum_{t=1}^T \gamma^{t-1} r_t | \pi]$, and $\rho^\pi$ is the state-action visitation distribution under $\pi$ from step $t = 1$ onwards.*

Note that the problem formulation here is different than the typical OPS/OPE setup used in existing works (Jiang & Li, 2016; Thomas & Brunskill, 2016; Doroudi et al., 2017; Yang et al., 2020; Zhang et al., 2021; Gao et al., 2022; Le et al., 2019), as only the initial state $s_0$ is available for policy selection. Such a formulation is aligned with use cases under HCSs, *e.g.*, treatment plan needs to be laid out soon after a new patient is admitted to the intensive care unit (ICU) in medical centers. However, most indirect OPS methods such as importance sampling (IS) (Precup, 2000; Doroudi et al., 2017) and doubly robust (DR) (Jiang & Li, 2016; Thomas & Brunskill, 2016) require the entire trajectory to be observed, in order to estimate $V^\pi$. Though direct methods like fitted-Q evaluation (FQE) (Le et al., 2019) could be used as a workaround, they do not take into account the unique characteristics for each participant that plays a crucial role in HCS applications; results in Section 3 show that they in general underperform in the real-world IE experiment. To address both challenges, we introduce the UOPS approach, starting with the sub-group partitioning step introduced below.

## 2.2 SUB-GROUP PARTITIONING

In this sub-section, we introduce the sub-group partitioning step that partition the participants in the offline dataset into sub-groups. Furthermore, value functions over all candidate policies $\pi \in \boldsymbol{\Pi}$ are learned respectively for each sub-group, to be leveraged as the OPS criteria for each sub-group.

**Sub-group partitioning.** The partitioning is performed over the initial state of each trajectory in the offline dataset, $\tau_\beta \in \mathcal{D}$. Given assumptions 1 and 2, and the fact that $\mathcal{S}_0(i)$'s in general only share limited support across participants (*i.e.*, every human has unique characteristics and past experience), such partitioning is essentially performed at per-participant level. Specifically, we

---

[1]Without loss of generality, in the rest of the paper, we use $\mathcal{S}_0$ to represent the marginal distribution of the initial states over all participants, while $\mathcal{S}_0(i)$ represents the distribution specific to the $i$-th participant.

consider partitioning the participants into $M$ sub-groups. Then for all sub-groups, $K_m$'s, in the set of sub-groups, $\mathcal{K} = \{K_1, \ldots, K_M\}$, we have $\bigcup_{m=1}^{M} K_m = \mathcal{S}_0$ and $K_m \cap K_n = \emptyset, \forall m \neq n$. The total number of groups $M$ needed can be determined using silhouette scores (Hallac et al., 2017). Denote the partition function $k(\cdot): \mathcal{S}_0 \to \mathcal{K}$. We then define the value function specific to each sub-group.

**Definition 1** (Value Function per Sub-group). *The value function over policy $\pi$, $V_{K_m}^{\pi}$, specific to the sub-group $K_m$, is the expected accumulative return over the initial states that correspond to the set of participants $\mathcal{I}_m = \{i | k(s_0^{(i)}) = K_m, i \in \mathcal{I}\}$ residing in the same sub-group. Specifically, $V_{K_m}^{\pi} = \mathbb{E}_{s_0 \sim Unif(\{\mathcal{S}_0(i)|i \in \mathcal{I}_m\}), (s_{t>0}, a_{t>0}) \sim \rho^{\pi}, r \sim R}[\sum_{t=1}^{T} \gamma^{t-1} r_t | \pi]$, with $s_0 \sim Unif(\{\mathcal{S}_0(i)|i \in \mathcal{I}_m\})$ representing that $s_0$ is sampled from a uniformly weighted mixture of distributions over $\{\mathcal{S}_0(i)|i \in \mathcal{I}_m\}$, pertaining to sub-group $K_m$.*

The goal of sub-group partitioning is to learn the partition function $k(\cdot)$, such that the difference between the value of the best policy candidate, $\max_{\pi \in \mathbf{\Pi}} V_{K_m}^{\pi}$, and the value of the behavioral policy, $V_{K_m}^{\beta}$, is maximized for all participants $i \in \mathcal{I}$ and sub-groups $K_m \in \mathcal{K}$, *i.e.*,

$$\max_k \sum_{i \in \mathcal{I}} \left[ \left( \max_{\pi \in \mathbf{\Pi}} V_{K_m=k\left(s_0^{(i)}\right)}^{\pi} \right) - V_{K_m=k\left(s_0^{(i)}\right)}^{\beta} \right]. \tag{1}$$

The objective (1) is designed in the sense that participants may benefit more from the type of policies that fit better for their individual characteristics. For example, in IE, different candidate lecturing policies may be used toward prospective high- and low-performers respectively, as justified by the findings from our real-world IE experiment (centered around Figure 2 in Section 3.2). The value provided by different policies for a specific type of learners (*i.e.*, sub-group) could be different, measured by $V_{K_m}^{\pi} - V_{K_m}^{\beta}$ for all $\pi \in \mathbf{\Pi}$; here, $V_{K_m}^{\beta}$ captures the expected return from a instructor-designed, one-size-fit-all baseline (*i.e.*, behavioral) policy that is used to collect offline data (Mandel et al., 2014; VanLehn, 2006; Zhou et al., 2022). Then, it would be crucial to identify to which group each student belongs, as it can maximize the returns collected by each student throughout the horizon.

**Optimization over the sub-typing objective** (1). The overall sub-typing objective (1) can be achieved using a two-step approach, *i.e.*, (*i*) *pre-partitioning* with offline dataset, followed by (*ii*) *deployment* upon observation of the initial states of arriving participants. Due to space limitation, the specific steps can be found in Appendix B.1.

**Theorem 1.** *Define the estimator $\hat{D}_{K_m}^{\pi,\beta}$ as, i.e.,*

$$\hat{D}_{K_m}^{\pi,\beta} = \frac{1}{|\mathcal{I}_m|} \sum_{i \in \mathcal{I}_m} \left( \omega_i \sum_{t=1}^{T} \gamma^{t-1} r_t^{(i)} - \sum_{t=1}^{T} \gamma^{t-1} r_t^{(i)} \right); \tag{2}$$

*here, $\mathcal{I}_m$ follows the definition above, which is the set of participants grouped in $K_m$; $\omega_i = \Pi_{t=1}^{T} \pi(a_t^{(i)}|s_t^{(i)})/\beta(a_t^{(i)}|s_t^{(i)})$ is the IS weight for the $i$-th trajectory in the offline dataset; $s_t^{(i)}, a_t^{(i)}, r_t^{(i)}$ are the states, actions, rewards logged in the offline trajectory, respectively. Then, $\hat{D}_{K_m}^{\pi,\beta}$ is unbiased, with its variance bounded by, i.e.,*

$$Var(\hat{D}^{\beta,\pi}) \leq \left\| \sum_{t=1}^{T} \gamma^{t-1} r_t \right\|_{\infty}^{2} \left( \frac{1}{ESS} - \frac{1}{N} \right), \tag{3}$$

*with $ESS$ being the effective sample size (Kong, 1992).*

The proof of theorem 1 is provided in Appendix B.2.

### 2.3 TRAJECTORIES AUGMENTATION WITHIN EACH SUB-GROUP

In HCSs, each sub-group may only contain a limited number of participants, due to the high cost of recruiting participants as well as time constraints in real-world experiments. For example, in the IE experiment in Section 3, one sub-group only contains 45 students as a result from sub-group partitioning. Consequently, the overall offline trajectories within each group may cover limited visitations of the state and action spaces, and make the downstream policy selection task challenging (Nie et al., 2022). Latent-model-based data augmentation has been commonly employed

in previous offline RL (Hafner et al., 2020; Lee et al., 2020; Rybkin et al., 2021; Gao et al., 2022), to resolve similar issues. For this sub-section, we specifically consider the variational auto-encoder (VAE) architecture introduced in Gao et al. (2022), as it is originally designed for offline setup as well. Now we briefly introduce the VAE setup, which can capture the underlying dynamics and generate synthetic offline trajectories to improve the state-action visitation coverage *within each subgroup*. Specifically, given the offline trajectories $\mathcal{T}_m$ specific to the subgroup $K_m$, the VAE consists of three major components, *i.e.*, $(i)$ the latent prior $p(z_0)$ that represents the distribution of the initial latent states over $\mathcal{T}_m$; $(ii)$ the encoder $q_\eta(z_t|s_{t-1}, a_{t-1}, s_t)$ that encodes the MDP transitions into the latent space; $(iii)$ the decoders $p_\xi(z_t|z_{t-1}, a_{t-1})$, $p_\xi(s_t|z_t)$, $p_\xi(r_{t-1}|z_t)$ that reconstructs new samples. The training objective is formulated as an evidence lower bound (ELBO) specifically derived for the architecture above. More details can be found in Appendix B.3. Consequently, for the trajectories in each subgroup, $\mathcal{T}_m$, the VAE can be trained to generate a set of synthetic samples, denoted as $\widehat{\mathcal{T}}_m$. In the Section 3.2, we further discuss and justify the need of trajectory augmentation through an real-world intelligent education (IE) experiment.

## 2.4 THE UOPS ALGORITHM

The overall flow of the UOPS framework is described in Algorithm 1. The training phase directly follow from the sub-sections above. Upon deployment, UOPS can help HCSs monitor each arriving participant, determine the sub-group the participant falls within, and select the policy to be deployed according to the initial state. Such real-time adaptability is important for HCSs in practice, and is different from existing OPS works which in general assume either the full trajectories or population characteristics are known (Keramati et al., 2022; Yang et al., 2022; Zhong et al., 2022). For example, in practical IE, students may start learning irregularly according to their own schedules, hence can create discrepancies in their start times. Such methods fall short in cases when selecting policies based on population or sub-group information in the

---

**Algorithm 1** UOPS.

**Input:** A set of target policies $\mathbf{\Pi}$, offline dataset $\mathcal{D}$.
**Begin:**
    // Training Phase.
1: Calculate the number of subgroups $M$ needed for $\mathcal{D}$, using silhouette scores (Hallac et al., 2017).
2: Obtain the sub-group partitioning $\mathbf{K} = \{K_1, \ldots, K_M\}$ following Section 2.2.
3: **for** each sub-group $K_m$ **do**
4:     Augment sub-group samples $\mathcal{T}_m$ with $\widehat{\mathcal{T}}_m$.
5:     Use the estimator in Theorem 1 to obtain $\hat{D}_{K_m}^{\pi,\beta}$ for all candidate target policies $\pi \in \mathbf{\Pi}$, over $\mathcal{T}_m \cup \widehat{\mathcal{T}}_m$.
6:     Select the best candidate target policy $\pi_m^*$ that maximizes $\hat{D}_{K_m}^{\pi,\beta}$ as the one to be deployed over $K_m$.
7: **end for**
    // Deployment Phase.
8: **while** the HCS receives the initial state $s_0$ from a new participant **do**
9:     Determine the sub-group $K_m$ for the new participant.
10:     Deploy to the participant the best candidate policy $\pi_m^*$ specific to sub-group $K_m$.
11: **end while**

---

upcoming semester – they requires the data from all arriving students are collected upfront, which would be unrealistic. Note that, to the best of our knowledge, we are the first work that formally consider the problem of sub-typing arriving participants, and UOPS is the first approach that solves this practical problem by introducing a framework that can work with HCSs in the real-world.

## 3 EXPERIMENTS

UOPS is tested over two types of HCSs, *i.e.*, intelligent education (IE) and healthcare. Specifically, the ***real-world IE experiment*** involves 1,148 student participating in college entry-level probability course across 6 academic semesters. The goal is to use the data collected from the students of the first 5 semesters, to assign pre-trained RL lecturing policies to every student enrolled in the 6-th semester, in order to maximize their learning outcomes. The healthcare experiment targets for selecting pre-configured policies that can best treat patients with sepsis, over a simulated environment widely adopted in existing works (Hao et al., 2021; Nie et al., 2022; Tang & Wiens, 2021; Lorberbom et al., 2021; Gao et al., 2023; Namkoong et al., 2020).

### 3.1 BASELINES

**Existing OPS/OPE.** The most straightforward approach to facilitate OPS in HCSs is to select policies via existing OPS/OPE methods, by choosing the candidate target policy $\pi \in \mathbf{\Pi}$ that achieves the maximum estimated return *over the entire offline dataset, i.e.*, indiscriminately across all potential sub-groups. Specifically, 6 commonly used OPE methods are considered, *i.e.*, Weighted IS (WIS) (Precup, 2000), Per-Decision IS (PDIS) (Precup, 2000), Fitted-Q Evaluation (FQE) (Le et al., 2019), Weighted

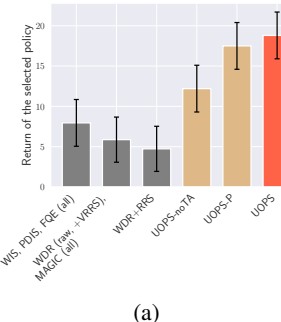 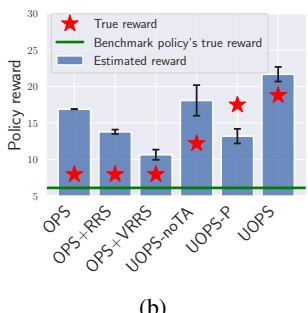

(a)  (b)

Figure 1: Analysis of main results from the real-world IE experiment. (a) Overall performance of the 6-th semester's student cohort. Methods that selected the same policy are merged in one bin, *i.e.*, `all` refers to all three variations (`raw, +RRS, +VRRS`) of the existing OPS baselines. (b) Estimated and true policy performance using each method. For *OPE*, *OPE+RRS*, *OPE+VRRS*, results with the least gap between estimated and true rewards among OPE methods (*i.e.*, WIS, FQE+RRS, and FQE+VRRS, respectively) are shown in the figure. True reward refers to the returns averaged over the cohort of the 6-th semester, obtained by deploying the policy selected for each student correspondingly.

DR (WDR) (Thomas & Brunskill, 2016), MAGIC (Thomas & Brunskill, 2016), and Dual stationary DIstribution Correction Estimation (DualDICE) (Nachum et al., 2019).

**Existing OPS/OPE with vanilla repeated random sampling (OPS+RRS).** We also compare UOPS against a classic data augmentation method in order to evaluate the necessity of the VAE-based method introduced in Section 2.3 – *i.e.*, repeated random sampling (RRS) with replacement of the historical data to perform OPE. RSS has shown superior performance in some human-related tasks, such as disease treatment (Nie et al., 2022). Specifically, all OPS/OPE methods considered above are applied to the RRS-augmented offline dataset, where the value of each candidate target policy is obtained by averaging over 20 sampling repetitions. However, note that RRS does not intrinsically consider the temporal relations among state-action transitions as captured by MDP.

**Existing OPS/OPE with VAE-based RRS (OPS+VRRS).** This baseline perform OPS with RRS on augmented samples resulted from the VAE introduced in Section 2.3, in order to allow RRS to consider MDP-typed transitions, hence improve state-action visitation coverage of the augmented dataset. This method can, to some extent, be interpreted as an ablation baseline of UOPS, by removing the sub-group partitioning step (Section 2.2), and slightly tweaking the VAE-based offline dataset augmentation step (Section 2.3) such that it does not need any sub-group information. Specifically, we set the amount of augmented data identical to the amount of original historical data, *i.e.*, $|\widehat{\mathcal{T}}| = |\mathcal{T}| = N$, and RRS $N$ samples from both set $\widehat{\mathcal{T}} \cup \mathcal{T}$ to perform OPE. Final estimates are averaged results from 20 repeated sampling processes.

**UOPS without trajectory augmentation (UOPS-noTA).** This is the ablation baseline that completely removes from UOPS the augmentation technique introduced in Section 2.3.

**UOPS for the population (UOPS-P).** We consider on additional ablation baseline that follows the same training steps as UOPS (*i.e.*, steps 1-7 of Alg. 1), but rather select *a single policy* that is identified (by UOPS) as the best for majority of the sub-groups, to be deployed to all participants. In other words, after training, UOPS produces the mapping $h : \mathcal{K} \to \mathbf{\Pi}$, while UOPS-P will always deploy to every arriving participant the policy that appears most frequently in the set $\{h(K_m)|K_m \in \mathcal{K}\}$.

## 3.2 THE REAL-WORLD IE EXPERIMENT

The IE system has been integrated into a undergraduate-level introduction to probability and statistics course over 6 semesters, including a total of 1,288 student participants. This study has received approval from the Institutional Review Board (IRB) at the institution to ensure ethical compliance. Additionally, oversight is provided by a departmental committee, which is responsible for safeguarding the academic performance and privacy of the participants. In this educational context, each learning session revolves around a student's engagement with a set of 12 problems, with this period referred to as an "episode" (horizon $T = 12$). During each step, the IE system offers students three

actions: independent work, utilizing hints, or directly receiving the complete solution (primarily for study purposes). The states space is constituted by by 140 features that have been meticulously extracted from the interaction logs by domain experts, which encompass various aspects of the students' activities, such as the time spent on each problem and the accuracy of their solutions. The learning outcome is issued as the environmental reward at the end of each episode (0 reward for all other steps), measured by the normalized learning gain (NLG) quantified using the scores received from two exams, *i.e.*, one taken before the student start using the system, and another after. Data collected from the first 5 semesters (over a lecturer-designed behavioral policy) are used to train UOPS for selecting from a set of candidate policies to be deployed to each student in the cohort of the 6-th semester, including 3 pre-trained RL policies and 1 benchmark policy (whose performance benchmark the lower-bound of what could be tested with student participants). See Appendix A for the definition of NLG, details on pre-trained RL policies, and more.

**Main Results.** Figure 1(a) presents students' performance under policies selected by different methods. Overall, UOPS was the most effective policy selection leading to the greatest average student performance. The return difference between UOPS and the two ablation, UOPS-noTA and UOPS-P, illustrate the importance of augmenting offline trajectories (as introduced in Section 2.3) and assign to arriving students policies that better fit the characteristics shared within their sub-groups, respectively. Moreover, most existing OPS/OPE methods tend to select sub-optimal policies that resulted in better learning gain than the benchmark policy. Note that we also observed that DualDICE could not distinguish the returns over all target policies; thus, it is unable to be used for policy selection in this experiment and we omit its results. It is also important to evaluate how accurate the value estimation $V^{\pi^*}$ would be for the best candidate policy selected across all methods, over the arriving student cohort at the 6-th semester, as illustrated in Figure 1(b). UOPS provideed more accurate policy estimation by achieving the smallest error between true and estimated policy rewards. With VRRS, most OPS methods improved their policy estimation performance, which was benefited from the richer state-action visitation coverage provided by the synthetic samples generated by VRRS. However, even with such augmentations, existing OPS methods still chose sub-optimal policies, which justified the importance of considering participant-specific characteristics in HCSs, which is tackled by sub-group partitioning in UOPS (Section 2.2).

**How does UOPS perform within each student sub-group?**
For a more comprehensive understanding of student behaviors affected by the policy being deployed in IE, we further investigate how the sub-groups are partitioned and how the policies being assigned to each sub-group perform. Specifically, UOPS identified four subgroups (*i.e.*, $K_1, K_2, K_3, K_4$) as a result of Section 2.2. Under the behavioral policy, the average NLG across all students is 0.9 with slight improvement after tutoring. Specifically, $K_1(N_{train} = 345, N_{test} = 30)$ and $K_2(N_{train} = 678, N_{test} = 92)$ achieved average NLG of 1.9 *[95% CI, 1.7, 2.1]*[2] and 0.7 *[95% CI, 0.6, 0.8]* under the behavioral policy, respectively. In the testing (6-th) semester, UOPS constantly selected the best performing policy for students identified as sub-groups $K_1$ and $K_2$, with learning outcomes improvement quantified as 17.7 *[95% CI, -1.7, 37.1]* and 13.6

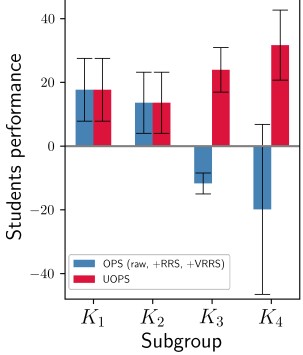

Figure 2: Performance of students over all four sub-groups under selected policies in the 6-th semester.

*[95% CI, -5.2, 32.4]* in terms of NLGs, respectively, as shown in Figure 2. This is on the same level achieved by the best possible baseline combinations worked for each student, regardless of the base OPS algorithm used, *i.e.*, the union of the best performance reported from the 18 baselines involving existing OPS methods (introduced in the first 3 paragraphs of Section 3.1) over each student. However, note that in reality one does not have access to such an oracle in terms of which 1 out of the 18 baseline methods would work well for each arriving student upfront (*i.e.*, at the beginning of the semester). In contrast, *UOPS achieved the same level of performance without the need for an oracle*. Note that sub-groups $K_1$ and $K_2$, were less sensitive to target policies and achieved positive NLG in both training and testing semester. On the other hand, offline data over the behavioral policy showed that $K_3(N_{train} = 101, N_{test} = 12)$ and $K_4(N_{train} = 24, N_{test} = 6)$ are associated with negative average NLGs of -0.5 *[95% CI, 1.7, 2.1]* and -1.5 *[95% CI, -0.3, 1.2]*, respectively, which can be considered low performers. It is observed that students in sub-group $K_3$ performed kept moving

---

[2]CI stands for confidence interval.

rapidly among questions while working on the IE system, indicating that they were not seriously tackling any one of the questions; while participants in $K_4$ abused hints, but still made much more mistakes in the meantime. Figure 2 also presents the NLG of students from the low-performing subgroups $K_3$ and $K_4$ under policies selected by the best existing-OPS baselines (following the oracle as above) and UOPS. Under UOPS, both subgroups achieved significant improvement (average NLGs 24.0 *[95% CI, 10.2, 37.7]* and 31.7 *[95% CI, 10.1, 53.3]*, respectively) compared to students in historical semesters. However, the sub-optimal policy chosen by baselines had a negative effect on both sub-groups (average NLGs -11.7 *[95% CI, -18.1, -5.3]* and -19.9 *[95% CI, -72.1, 32.4]*, respectively); see Figure 1(a). Such an observation particularly justifies the need for personalizing the policies deployed to different type of participants (*i.e.*, students), especially for the sub-groups (*i.e.*, low-performers), since they can be more sensitive to policy selection. Based on the statistics reported above, UOPS improved the NLG of students by 208% over the lecturer-designed behavioral policy, and by 136% over the union of the best performance achieved across existing-OPS-based baselines. To this end, the UOPS framework has the potential to facilitate fairness in RL-empowered HCSs in general – we have discussed this in details in Appendix A.5.

### 3.3 THE HEALTHCARE EXPERIMENT

In this experiment, we consider selecting the policy that can best treat sepsis for each patient in the ICU, leveraging the simulated environment introduced by Oberst & Sontag (2019), which has been widely adopted in existing works (Hao et al., 2021; Nie et al., 2022; Tang & Wiens, 2021; Lorberbom et al., 2021; Gao et al., 2023; Namkoong et al., 2020). Specifically, the state space is constituted by a binary indicator for diabetes, and four vital signs {heart rate, blood pressure, oxygen concentration, glucose level} that take values in a subset of {very high, high, normal, low, very low}; size of the state space is $|\mathcal{S}| = 1440$. Actions are captured by combinations of three binary treamtment options, {antibiotics, vasopressors, mechanical ventilation}, which lead to $|\mathcal{A}| = 2^3$. Following Namkoong et al. (2020), three candidate target policies are considered, *i.e.*, $(i)$ without antibiotics (WOA) which does not administer antibiotics right after the patient is admitted, $(ii)$ with antibiotics (WA) that always administer antibiotics once the patient is admitted, $(iii)$ an RL policy trained following policy iteration (PI). Note that as pointed by Namkoong et al. (2020), the true returns of WA and PI are usually close, since antibiotics are in general helpful for treating sepsis, which is also observed in our experiment; see Table 1. Moreover, a simulated unrecorded comorbidities is applied to the cohort, capturing the uncertainties caused by patient's underlying diseases (or other characteristics), which could reduce the effects of the antibiotics being administered. See Appendix C.4 for more details in regards to the environmental setup.

Given the simulated environment, we mainly consider using this experiment to evaluate the source of improvement brought in by the sub-group partitioning step (Section 2.2) in UOPS. Specifically, multiple scaled offline datasets are generated, representing different degrees of the state-action visitation coverage – we vary the total number of trajectory $N$={2,500, 5,000, 10,000}, in lieu of performing trajectory augmentations for both UOPS and existing OPS baselines. In other words, in this experiment, we consider the UOPS without the VAE augmentation step introduced in Section 2.3, as well as the 6 original OPS baselines (without any RRS/VRRS) introduced in Section 3.1. We believe this setup would help isolate the source of improvements brought in by sub-group partitioning. The average absolute errors (AEs), in terms of OPE, and returns, in terms of OPS, resulted from deploying to each patient the corresponding candidate policy selected by UOPS against baselines, are reported in Table 1. It can be observed that UOPS achieved the lowest AE and highest return regardless of the size of the offline dataset. We additionally evaluate the top-1 regret (*i.e.*, regret@1) of the selected policy following UOPS and baselines, which are also reported in Table 1. It can be observed that UOPS achieved exceedingly low regrets compared to baselines. Both observations emphasize the effectiveness of the sub-group partitioning technique leveraged by UOPS, as the environment does capture comorbidities as part of the participant characteristics. Moreover, the AEs and regrets of most methods decrease when the size of offline dataset increase, justifying that improved state-action visitation coverage provided by the offline trajectories is crucial for reducing estimation errors and improving policy selection outcomes (*i.e.*, the motivation of trajectory augmentation introduced in Section 2.3).

## 4 RELATED WORKS

**Off-policy selection (OPS).** OPS are typically approached via OPE in existing works, by estimating the expected return of target policies using historical data collected under a behavior policy. A

Table 1: The absolute errors (AEs) and returns resulted from deploying to each patient the corresponding candidate policy selected by UOPS against baselines, as well as the top-1 regret (regret@1) of the selected policy, averaged over 10 different simulation runs. Standard errors are rounded.

|  |  | UOPS | WIS | PDIS | FQE | WDR | MAGIC |
|---|---|---|---|---|---|---|---|
| N=2,500 | AE | **0.026±0.00** | 0.054±0.00 | 0.109±0.00 | 0.070±0.01 | 8.281±0.00 | 5.681±0.00 |
|  | Return | **0.132±0.02** | 0.121±0.01 | 0.121±0.01 | 0.129±0.00 | 0.121±0.01 | 0.129±0.00 |
|  | Regret@1 | **0.042±0.01** | 0.066±0.00 | 0.066±0.00 | 0.106±0.09 | 0.066±0.00 | 0.106±0.09 |
| N=5,000 | AE | **0.006±0.00** | 0.046±0.00 | 0.082±0.00 | 0.073±0.01 | 3.443±0.01 | 3.238±0.01 |
|  | Return | **0.149±0.01** | 0.123±0.01 | 0.123±0.01 | 0.121±0.01 | 0.123±0.01 | 0.123±0.01 |
|  | Regret@1 | **0.020±0.00** | 0.050±0.00 | 0.050±0.00 | 0.208±0.13 | 0.050±0.00 | 0.050±0.00 |
| N=10,000 | AE | **0.022±0.00** | **0.022±0.00** | 0.097±0.00 | 0.105±0.00 | 0.995±0.01 | 1.210±0.01 |
|  | Return | **0.130±0.00** | 0.129±0.00 | 0.121±0.00 | 0.121±0.00 | 0.121±0.00 | 0.121±0.00 |
|  | Regret@1 | **0.016±0.00** | 0.019±0.00 | 0.029±0.01 | 0.029±0.01 | 0.029±0.01 | 0.029±0.01 |

variety of contemporary OPE methods has been proposed, which can be mainly divided into three categories (Voloshin et al., 2021b): (*i*) direct methods that directly estimate the value functions of the evaluation policy (Nachum et al., 2019; Uehara et al., 2020; Zhang et al., 2021; Yang et al., 2022), including but not limited to model-based estimators (MB) (Paduraru, 2013; Zhang et al., 2021), value-based estimators (Le et al., 2019) such as Fitted Q Evaluation (FQE), and minimax estimators (Liu et al., 2018; Zhang et al., 2020; Voloshin et al., 2021a) such as DualDICE (Yang et al., 2020); (*ii*) inverse propensity scoring, or indirect methods (Precup, 2000; Doroudi et al., 2017), such as Importance Sampling (IS) (Doroudi et al., 2017); (*iii*) hybrid methods combine aspects of both inverse propensity scoring and direct methods (Jiang & Li, 2016; Thomas & Brunskill, 2016), such as DR (Jiang & Li, 2016). In practice, due to expensive online evaluations, researchers generally selected the policy with the highest estimated rewards via OPE. For example, Mandel et al. selected the policy with the maximum IS score to be deployed to an educational game (Mandel et al., 2014). Recently, some works focused on estimator selection or hyperparameter tuning in off-policy selection (Nie et al., 2022; Zhang & Jiang, 2021; Xie & Jiang, 2021; Su et al., 2020; Miyaguchi, 2022; Kumar et al., 2022; Lee et al., 2022; Tang & Wiens, 2021; Paine et al., 2020). However, retraining policies may not be feasible in HCSs as online data collection is time- and resource-consuming. More importantly, prior work generally selected policies without considering the characteristics of participants, while personalized policy is flavored towards the needs specific to HCSs.

**RL-empowered automation in HCSs.** In modern HCSs, RL has raised significant attention toward enhancing the experience of human participants. Previous studies have demonstrated that RL can induce IE policies (Shen & Chi, 2016; Mandel et al., 2014; Wang et al., 2017; Zhou et al., 2022; Sanz Ausin et al., 2020). For example, Zhou et al. (Zhou et al., 2022) applied hierarchical reinforcement learning (HRL) to improve students' normalized learning gain in a Discrete Mathematics course, and the HRL-induced policy was more effective than the Deep Q-Network induced policy. Similarly, in healthcare, RL has been used to synthesize policies that can adapt high-level treatment plans (Raghu et al., 2017; Namkoong et al., 2020; Lorberbom et al., 2021), or to control medical devices and surgical robotics from a more granular level (Gao et al., 2020; Lu et al., 2019; Richter et al., 2019). Since online evaluation/testing is high-stake in practical HCSs, effective OPS methods are important in closing the loop, by significantly reducing the resources needed for online testing/deployment and preemptively justifying safety of the policies subject to be deployed.

## 5 CONCLUSION AND LIMITATION

In this work, we introduced the UOPS framework that facilitated policy selection in HCSs; it tackled the heterogeneity of participants by sub-group partitioning. Unlike existing OPS methods, UOPS customized the policy selection criteria for each sub-group respectively. UOPS was tested in a real-world IE experiment and a simulated sepsis treatment environment, which significantly outperformed baselines. Though in the future it would be possible to extend UOPS to a offline RL policy optimization framework, however, in this work we specifically focus on the OPS task in order to isolate the source of improvements brought in by sub-group partitioning and trajectory augmentation. Future avenues along the line of UOPS also include deriving estimators (for Theorem 1) that allow bias-variance trade off, *e.g.*, by integrating WDR or MAGIC (to substitute the IS weights).

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

## A    Detailed Setup of the IE Experiment and Additional Discussions

### A.1    The IE system for the college entry-level probability course.

Though the problem setting and our method are general and can be applied to other interactive IE systems, we primarily focus on the system specifically used in an undergraduate probability course at a university, which has been extensively used by over $1,288$ students with $\sim$800k recorded interaction logs through 6 academic years. The IE system is designed to teach entry-level undergraduate students with ten major probability principles, including complement theorem, mutually exclusive theorem, independent events, De Morgan's theorem, addition theorem for two events, addition theorem for three events, conditional independent events, conditional probability, total probability theorem, and Bayes' rule.

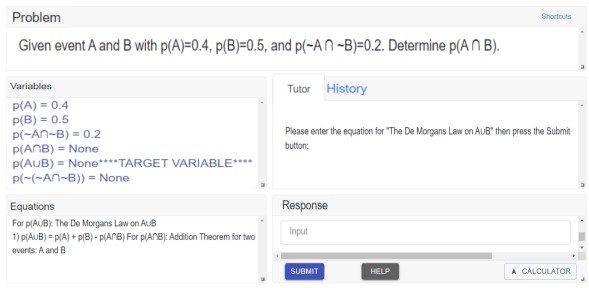

Each students went through four phases, including (*i*) reading the textbook; (*ii*) pre-exam; (*iii*) studying on the IE system; and (*iv*) post-exam. During the reading textbook phase, students read a general description of each principle, review examples, and solve some training problems to get familiar with the IE system. Subsequently, they take a pre-exam comprising a total of 14 single- and multiple-principle problems. During the pre-exam, students are not provided with feedback on their answers, nor are they allowed to go back to earlier questions (so as the post-exam). Then, students proceed to work on the IE system, where they receive the same 12 problems in a predetermined order. After that, students take the 20-problem post-exam, where 14 of the problems are isomorphic to the pre-exam and the remainders are non-isomorphic multiple-principle problems. Exams are auto-graded following the same grading criteria set by course instructors.

Figure 3: Graphical user interface (GUI) of the IE system. The problem statement window (*top*) presents the statement of the problem. The dialog window (*middle right*) shows the message the tutor provides to the students. Responses, e.g., writing an equation, are entered in the response window (*bottom right*). Any variables and equations generated through this process are shown on the variable window (*middle left*) and equation window (*bottom left*).

Since students' underlying characteristics and mind states are inherently unobservable (Mandel et al., 2014), the IE system defined its state space with 142 features that could possibly capture students' learning status based on their interaction logs, as suggested by domain experts. While tutoring, the agent makes decisions on two levels of granularity: problem-level first and then step-level. For problem-level, it first decides whether the next problem should be a worked example (`WE`) (Sweller & Cooper, 1985), problem-solving (`PS`), or a collaborative problem-solving worked example (`CPS`) (Schwonke et al., 2009). In `WE`s, students, observe how the tutor solves a problem; in `PS`s, students solve the problem themselves; in `CPS`s, the students and the tutor co-construct the solution. If a `CPS` is selected, the tutor will then make step-level decisions on whether to `elicit` the next step from the student or to `tell` the solution step to the student directly. Besides post-exam score, another important measure of student learning outcomes is their normalized learning gain (NLG), which is calculated by their pre- and post-exam scores $NLG = \frac{score_{postexam} - score_{preexam}}{\sqrt{1 - score_{preexam}}}$.

The NLG defined in (Chi et al., 2011), represents the extent to which students have benefited from the IE system in terms of improving their learning outcomes.

### A.2    Classroom Setup

**Participants recruitment.** All participants were entry-level undergraduates majoring in STEM and enrolled in the Probability course in a college. They were recruited via invitation emails and told the procedure of the study and their data were used for research purpose only, and the study was an opt-in without influence on their course grades. Participants can also opt-in not recording their logs and quit the study any time. No demographics data or course grades were collected. All participants had acknowledged the study procedure and future research conducted using their logs.

**Principles taught by the IE system.** Table 3 shows all ten principles for the IE system to teach designed for the undergraduate entry-level students with STEM majors.

**Pre- and post-exams.** As introduced in Section 2, we use pre- and post-exams to measure the extent to which students have benefited from the IE system for improved learning outcomes. Tables 4 & 5 contain all problems in pre- and post-exams during our experiment with the IE system.

**The set of candidate target policies under consideration.** For safety concerns, only three RL-induced target policies that passed expert sanity checks can be deployed, while the expert policy still remained in the semester as the control group. For fairness concerns, the IE system randomly assigned a policy to each student, while we tracked the policies selected by UOPS on each subgroup to evaluate its effectiveness. The chi-squared test was employed to check the relationship between policy assignment and subgroups, and it showed that the policy assignment cross subgroups were balanced with no significant relationship (p-value=0.479). In the testing semester, 140 students accomplished all problems and exams.

We provide the design of each problem regarding principles coverage for readers' interests. Detailed problem descriptions are omitted for identity and anonymity, which are only accessible within the research groups under IRB. An example problem description is shown in Figure 3.

## A.3 Environmental Setup of the IE system

**State Features.** The state features were defined by domain experts that could possible capture students' learning status based on their interaction logs. In sum, 142 features with both discrete and continuous values are extracted, we provide summary descriptions of the features characterized by their systematic functions: (*i*) Autonomy (10 features): the amount of work done by the student, such as the number of times the student restarted a problem; (*ii*) Temporal Situation (29 features): the time-related information about the work process, such as average time per step; (*iii*) Problem-Solving (35 features): information about the current problem-solving context, such as problem difficulty; (*iv*) Performance (57 features): information about the student's performance during problem-solving, such as percentage of correct entries; (*v*) Hints (11 features): information about the student's hint usage, such as the total number of hints requested.

**Actions & rewards.** See A.1 above.

**Behavior policy.** The behavior policy follows an expert policy commonly used in e-learning (Zhou et al., 2019), randomly taking the next problem as a worked example (WE), problem-solving by students (PS), or a collaborative problem-solving working examples (CPS). Note that the three decision choices are designed by domain experts that are found can support students' learning in prior works (Schwonke et al., 2009; Sweller & Cooper, 1985), thus the expert policy is considered as effective.

**Target (evaluation) policies.** The three RL-induced policies were trained using off-policy DQN algorithm with different hyper-parameter settings $lr = \{1e-3, 1e-4, 1e-5\}$, and passed expert sanity check. In this study, expert sanity check were conducted by departments and independent instructors.

## A.4 Illustrate Sub-group Partitioning with the IE Experiment

The initial logs (serving as the initial states in the MDP) of students are used for sub-group partitioning, which is now only benefited from the UOPS framework design but over two educational perspectives. First, initial logs may reflect not only the background knowledge of students but their interaction habits (Gao et al., 2021), without specific information related to behavior policies that may be distracting for sub-group partitioning. Though some existing works utilize demographics or grades of students from their prior taken courses to identify student subgroups (Castro-Wunsch et al., 2017; Sinclair et al., 1999), it may not be feasible in practice due to the protection of student information by institutions. Second, prior works have found that initial logs can be informative to indicate learning outcomes of students (Mao, 2019), which makes it possible for the IE system to customize the policies with the goal of improving learning outcomes for each subgroup.

However, there is a challenge with sub-group partitioning over the initial logs of students. The state space of student logs in the IE system is usually high-dimensional, due to the detailed capture of each

step taken during interaction and associated timing information (Chi et al., 2011; Mandel et al., 2014). For example, in this study, 142 features have been recorded. While some features might be irrelevant for downstream data mining tasks, it is challenging to determine their relevance a priori (Mandel et al., 2014). To solve this, we used a data-driven feature taxonomy over the state features of students, then perform subgroup partitioning with distilled features based on the feature taxonomy.

**The data-driven feature taxonomy over state features of students.** Educational researchers have used feature taxonomy in qualitative ways to support instructors subgroup students and understand behaviors of students (Marwan et al., 2020). Unlike prior approaches that are expensive requiring much effort from human experts, we used a data-driven feature taxonomy for a straightforward student subgroup partitioning that may reflect the knowledge background and dynamic learning progress of students. Specifically, we define three major categories of features according to their temporal and cross-student characteristics: (i) *System-assigned*: the features, which are static across students on the same problem, are assumed to be assigned by the system; (ii) *Student-centric*: the features, which differ across students from the initial logs and may change over time, is assumed to be students-centric and reflect both students' initial knowledge background and the changes of individual underlying mindset during learning; (iii) *Interaction-driven*: the features, which contain characteristics from both system-assigned and student-centric types, are assumed to be mixed-style features that are affected by both system and individuals. For example, the number of `tells` since `elicit` is set with a default value by the system while changing over time depending on students' progress.

**Sub-group partitioning with distilled features via feature taxonomy.** Since system-assigned features are mainly dominated by system design and remain static across students on each problem, for the purpose of subgroup partitioning, we focus on the two types of features, student-centric

Table 2: Feature taxonomy with examples and percentage in the the IE system.

| Taxonomy | Examples | Perc. |
|---|---|---|
| System-assigned | Problem difficulty | 18% |
| Student-centric | Number of hints requested | 48% |
| Interaction-driven | Number of `tells` since `elicit` | 34% |

and interaction-driven, since both could be highly associated with students' underlying mental status and behaviors, for which we call *student-sensitive* features. In this work, we identified $82\%(117)$ from overall 142 features as student-sensitive features and used them for subgroup partitioning. Specifically, to learn the subgroups, we leverage Toeplitz inverse covariance-based clustering (TICC) (Hallac et al., 2017) to map initial logs $\mathcal{S}_0$ into $M$ clusters based on the values of student-sensitive features, where each $s_0 \in \mathcal{S}_0$ is associated with a cluster from the set $\mathbf{K} = \{K_1, \ldots, K_M\}$. The initial logs that are mapped to the same cluster can be considered to share the graphical connectivity structure of cross-features information captured by TICC. We consider using TICC because of its superior performance in clustering compared to traditional distance-based methods such as K-means, especially with human behavior-related tasks (Hallac et al., 2017; Yang et al., 2021). The number of clusters can be determined by silhouette scores following (Hallac et al., 2017). Note that we exhibit TICC as an example in our proposed pipeline, while it can be replaced by other partitioning approaches if needed. Then, we assume subgroup partitioning is consistent with cluster assignments associated with initial logs, *i.e.*, students whose initial logs are associated with the same cluster index are considered from the same subgroup.

## A.5 MORE DISCUSSIONS OVER THE RESULTS FROM SECTION 3.2

**Would sub-group partitioning over a longer trajectory improve the performance of the OPS+VRRS baselines?** Recall that OPS+VRRS deployed the sub-optimal to most students, while their estimation accuracy (*i.e., absolute error*) was improved compared to purely OPS and OPS+RRS (Figure 1(b)), which is outperformed by UOPS over a slight margin. We further investigate the effects of subg-roup partitioning with longer trajectory information on OPS+VRRS performance. We conduct sub-group partitioning over the length of trajectories, *i.e.*, perform sub-group partitioning on the averaged states' features associated with the first $\Delta$ problems across historical trajectories, where $\Delta \in [1, 11]$ excluding the final problem. Then we augment the same amount of samples for each subgroup $K$, *i.e.*, $|\widehat{\mathcal{T}}_K| = |\mathcal{T}_K| = |K|$ and perform OPS+RRS. We observe that in all 55 conditions except the five (*i.e.*, WIS+VRRS $\Delta = 4, 11$, PDIS+VRRS $\Delta = 8$, and FQE+VRRS $\Delta = 7, 8$), all OPS+VRRS still select the sub-optimal policy. Figure 4 presents the mean absolute error (MAE) of

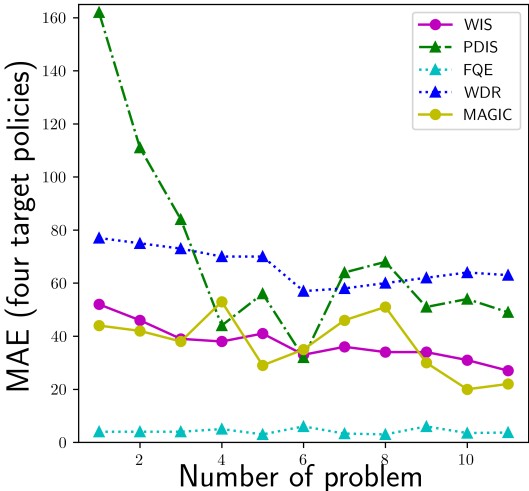

Figure 4: Mean absolute error (MAE) of OPE AugRRS with subgroup partitioning over problems in historical data.

the OPS+VRRS methods over the four target policies. It shows the trend of improved MAE over the number of problems for most methods. Those indicate that more information over a longer trajectory does have some positive effects on OPS+VRRS, but their policy selection is hard to be improved and stabilized. More students-centric and robust OPS methods are needed for IE policy selection.

**Broader impact on facilitating fairness in RL-empowered HCSs.** Fairness in AI-empowered HCSs has been a long-standing concern (Lepri et al., 2021; Metevier et al., 2019; Nie et al., 2023; Ruan et al., 2023; Elmalaki, 2021). The UOPS framework can be potentially extended to promote fairness to a certain extent, by helping minority/under-represented groups to boost their utility gain through deployment of customized policy specific to the group. Specifically, following UOPS, the sub-group partitioning step can identify the small-scaled yet important groups, then the policy that is most beneficial for the group can be deployed to maximize the gain. As illustrated by Figure 2, UOPS effectively identifies the low-performing students (group $K_4$), which only constitute $< 5\%$ of the overall population at the 6-th (testing) semester, without leveraging any subjective information *a priori* (i.e., sub-group partitioning uses exactly the same features across all students). UOPS then significantly boosts their performance by deploying the policy most suitable for the group. Similarly, one can easily extend the UOPS framework to intelligent HCSs oriented toward other applications, in order to identify the groups that potentially need more attention, and help all participants to achieve similar gain indiscriminately by deploying the right policy to each participant.

Table 3: Principles taught by the IE system for undergraduate entry-level students.

| Abbr. | Name of principle | Expression |
|---|---|---|
| CT | Complement Theorem | $P(A) + P(\neg A) = 1$ |
| MET | Mutually Exclusive Theorem | $P(A \cap B) = 0$ iff A and B are mutually exclusive events |
| IE | Independent Events | $P(A \cap B) = P(A)P(B)$ if A and B are independent events |
| DMT | De Morgan's Theorem | $P(\neg(A \cup B)) = P(\neg A \cap \neg B), P(\neg(A \cap B)) = P(\neg A \cup \neg B)$ |
| A2 | Addition Theorem for two events | $P(A \cup B) = P(A) + P(B) - P(A \cap B)$ |
| A3 | Addition Theorem for three events | $P(A \cup B \cup C) = P(A) + P(B) + P(C) - P(A \cap B) - P(A \cap C) - p(B \cap C) + P(A \cap B \cap C)$ |
| CIE | Conditional Independent Events | $P(A \cap B|C) = P(A|C)P(B|C)$ if A and B are independent events given C |
| CP | Conditional Probability | $P(A \cap B) = P(A|B)P(B) = P(B|A)P(A)$ |
| TPT | Total Probability Theorem | $P(A) = P(A|B_1)P(B_1) + P(A|B_2)P(B_2) + \ldots + P(A|B_n)P(B_n)$ if $B_1, B_2, \cdots, B_n$ are mutually exclusive events and $B_1 \cup B_2 \cup \cdots \cup B_n = W$ |
| BR | Bayes Rule | $P(B_i|A) = P(A|B_i)P(B_i)/P(A|B_1)P(B_1) + P(A|B_2)P(B_2) + \cdots + P(A|B_n)P(B_n)$ if $B_1, B_2, \cdots, B_n$ are mutually exclusive events and $B_1 \cup B_2 \cup \cdots \cup B_n = W$ |

Table 4: Pre-exam problems in the IE system.

| Problem | CT | MET | IE | DMT | A2 | A3 | CIE | CP | TPT | BR |
|---|---|---|---|---|---|---|---|---|---|---|
| 1 | | | | | | ✓ | | | | |
| 2 | | | | | | | | | ✓ | |
| 3 | | | | | | | | ✓ | | |
| 4 | ✓ | | | ✓ | ✓ | | | | | |
| 5 | | | | | | | | | | ✓ |
| 6 | | | | | | | ✓ | | | ✓ |
| 7 | | ✓ | ✓ | | | ✓ | | | | |
| 8 | ✓ | ✓ | | ✓ | ✓ | ✓ | | | | |
| 9 | ✓ | | | | | | | | | |
| 10 | | | | | | ✓ | | | | |
| 11 | | | ✓ | | | | | | | |
| 12 | | | | | ✓ | | | | | |
| 13 | | | | | | | ✓ | | | |
| 14 | | ✓ | | | | | | | | |

Table 5: Post-exam problems in the IE system.

| Problem | CT | MET | IE | DMT | A2 | A3 | CIE | CP | TPT | BR | Iso-Test-Problem |
|---|---|---|---|---|---|---|---|---|---|---|---|
| 1 | | | ✓ | | | | | | | | 11 |
| 2 | | ✓ | ✓ | | | ✓ | | | | | 7 |
| 3 | ✓ | | | ✓ | ✓ | | | | | | 4 |
| 4 | ✓ | | | | | | | | | | 9 |
| 5 | | | | | | | | ✓ | | | 3 |
| 6 | | | | | ✓ | | | | | | 10 |
| 7 | | | | | | | | | ✓ | | 2 |
| 8 | | | | | | | ✓ | | | | 13 |
| 9 | | | | | | | | ✓ | ✓ | ✓ | N/A |
| 10 | | ✓ | | | | | | | | | 14 |
| 11 | | | | | | | | | | ✓ | 5 |
| 12 | | | | ✓ | | | | | | | 12 |
| 13 | | | | | | | ✓ | | | ✓ | 6 |
| 14 | ✓ | | ✓ | | | | | | | | N/A |
| 15 | ✓ | | | ✓ | ✓ | | | ✓ | | | N/A |
| 16 | ✓ | | | | | | | | | ✓ | N/A |
| 17 | | | | | | ✓ | | | | | 1 |
| 18 | ✓ | ✓ | | ✓ | ✓ | ✓ | | | | | 8 |
| 19 | ✓ | ✓ | | | ✓ | ✓ | | | | | N/A |
| 20 | | ✓ | | | | ✓ | | | ✓ | | N/A |

# B MORE ON THE METHODOLOGY

## B.1 OPTIMIZATION OVER THE SUB-TYPING OBJECTIVE (1).

The overall sub-typing objective (1) can be achieved using a two-step approach, *i.e.*, (*i*) *pre-partitioning* with offline dataset, followed by (*ii*) *deployment* upon observation of the initial states of arriving participants.

To facilitate (*i*), existing time-series clustering methods, such as Toeplitz Inverse Covariance-Based Clustering (TICC) and its variants (Hallac et al., 2017; Yang et al., 2021), is used to learn a preliminary partitioning $l : \mathcal{S}_0 \to \mathcal{K}$. Then, the value function estimator $\hat{D}^{\pi,\beta}_{K_m=l(s_0)}$ is trained for estimating $V^{\pi}_{K_m=l(s_0)} - V^{\beta}_{K_m=l(s_0)}$ using part of the offline trajectories whose initial state $s_0$ falls in the corresponding group $K_m = l(s_0)$ following Theorem 1 below, for all $K_m \in \mathcal{K}$.[3] At last, one can learn a mapping $d : \mathcal{S}_0 \times \Pi \times \mathcal{K} \to \mathbb{R}$ to reconstruct $D^{\pi,\beta}_{K_m}$'s estimation using the $(s_0, \pi, K_m)$ triplets as the inputs (*e.g.*, by minimizing squared error).

In step (*ii*), the final partitioning $k(\cdot)$ is obtained by exhaustively iterating over all possible cases for each *arriving* participant $i' \geq N$. Specifically, plug into $d(s_0^{(i')}, \pi, K_m)$ all policy candidates $\pi \in \Pi$ and all possible partitions $K_m \in \mathcal{K}$. Then, one can determine which partition $K_m$ satisfies $\max_{\pi \in \Pi} d(s_0^{(i')}, \pi, K_m)$. Then, assign to participant $i'$ the corresponding group $k(s_0^{(i')}) = K_m$, as captured by the mapping function $k : \mathcal{S}_0 \to \mathcal{K}$.

## B.2 PROOF OF BOUND 3

**Rényi divergence.** For $\epsilon \geq 0$, the Rényi divergence for two distribution $\pi$ and $\beta$ is defined by (Rényi, 1961)

$$d_\epsilon(\pi \| \beta) = \frac{1}{\epsilon - 1} log_2 \sum_x \beta(x) \Big( \frac{\pi(x)}{\beta(x)} \Big)^{\epsilon - 1}.$$

(Cortes et al., 2010) denote the exponential in base 2 by $d_\epsilon(\pi \| \beta) = 2^{D_\epsilon(\pi \| \beta)}$.

*Proof.* $\hat{D}^{\pi,\beta}_{K_m} = \frac{1}{|\mathcal{I}_m|} \sum_{i \in \mathcal{I}_m} \Big( \omega_i \sum_{t=1}^{T} \gamma^{t-1} r_t^{(i)} - \sum_{t=1}^{T} \gamma^{t-1} r_t^{(i)} \Big)$, with $\omega_i = \Pi_{t=1}^{T} \pi(a_t^{(i)} | s_t^{(i)}) / \beta(a_t^{(i)} | s_t^{(i)})$, can be upper bounded by the variance of importance sampling weights. Denote $g_i = \sum_{t=1}^{T} \gamma^{t-1} r_t^{(i)}$. Following (Keramati et al., 2022), since $Var(\hat{G}) \leq \mathbb{E}[\hat{G}^2]$ (following the definition of variance),

$$Var(\hat{D}^{\pi,\beta}_{K_m}) \leq \frac{\|g\|_\infty^2}{N^2} \mathbb{E}\Big[ \sum_i (\omega_i - 1)^2 \Big]$$

$$= \frac{1}{N} \|g\|_\infty^2 Var(\omega),$$

with the last equality following the fact $\mathbb{E}[\omega] = 1$. Moreover, $Var(w) = d_2(\pi \| \beta) - 1$ as pointed out by (Cortes et al., 2010), following the Rényi divergence (Rényi, 1961). Thus, the variance of the estimator $Var(\hat{D}^{\pi,\beta}_{K_m})$ is:

$$Var(\hat{D}^{\pi,\beta}_{K_m}) \leq \|g\|_\infty^2 \Big( \frac{d_2(\pi \| \beta) - 1}{N} \Big)$$

$$= \|g\|_\infty^2 \Big( \frac{d_2(\pi \| \beta)}{N} - \frac{1}{N} \Big).$$

The expression can be related to the effective sample size (ESS) of the original data given the target policy (Metelli et al., 2018), resulting in

$$Var(\hat{D}^{\pi,\beta}_{K_m}) \leq \|g\|_\infty^2 \Big( \frac{1}{ESS} - \frac{1}{N} \Big),$$

which completes the proof. □

---

[3] Note that $\hat{D}^{\pi,\beta}_{K_m=l(s_0)}$ essentially approximates the sum of value difference in (1) over each $\mathcal{I}_m$.

**Remark.** Note that in the special case that behavior policy being the same as target policy, the bound evaluates to zero. Moreover, as noted by (Keramati et al., 2022), denote the right-hand side of inequality 3 by $Var_u(\cdot)$, it can be used in each sub-group as a proxy of variance of the estimator in the subgroup, i.e.,

$$Var_u(\hat{D}_{K_m}^{\pi,\beta}) = \|g_m\|_\infty^2 \big( \frac{1}{ESS(K_m)} - \frac{1}{|K_m|} \big);$$

here, $g_m$ refers to the total return of the trajectories pertaining to the sub-group $K_m$, and $ESS(K_m)$ can be estimated by $ESS$ using $\widehat{ESS}(K_m) = \frac{(\sum_{i \in \mathcal{I}_m} g_i)^2}{\sum_{i \in \mathcal{I}_m} g_i^2}$ (Owen, 2013).

### B.3 DETAILED FORMULATION OF VAE IN MDP

**The latent prior** $p(z_0) \sim \mathcal{N}(0, I)$ representing the distribution of the initial latent states (at the beginning of each PST in the set $\mathcal{T}^g$), where $I$ is the identity covariance matrix.

**Encoder.** $q_\eta(z_t|s_{t-1}, a_{t-1}, s_t)$ is used to approximate the posterior distribution $p_\xi(z_t|s_{t-1}, a_{t-1}, s_t) = \frac{p_\xi(z_{t-1}, a_{t-1}, z_t, s_t)}{\int_{z_t \in \mathcal{Z}} p(z_{t-1}, a_{t-1}, z_t, s_t) dz_t}$, where $\mathcal{Z} \subset \mathbb{R}^m$ and $m$ is the dimension. Given that $q_\eta(z_{0:T}|s_{0:T}, a_{0:T-1}) = q_\eta(z_0|s_0) \prod_{t=1}^T q_\eta(z_t|z_{t-1}, a_{t-1}, s_t)$, both distributions $q_\eta(z_0|s_0)$ and $q_\eta(z_t|z_{t-1}, a_{t-1}, s_t)$ follow diagonal Gaussian, where mean and diagonal covariance are determined by multi-layer perceptrons (MLPs) and long short-term memory (LSTM), with neural network weights $\eta$. Thus, one can infer $z_0^\eta \sim q_\eta(z_0|s_0)$, $z_t^\eta \sim q_\eta(z_t|h_t^\eta)$, with $h_t^\eta = f_\eta(h_{t-1}^\eta, z_{t-1}^\eta, a_{t-1}, s_t)$ where $f_\eta$ represents LSTM layer and $h_t^\eta$ represents LSTM recurrent hidden state.

**Decoder.** $p_\xi(z_t, s_t, r_{t-1}|z_{t-1}, a_{t-1})$ is used to sample new trajectories. Given $p_\xi(z_{1:T}, s_{0:T}, r_{0:T-1}|z_0, \xi) = \prod_{t=0}^T p_\xi(s_t|z_t) \prod_{t=1}^T p_\xi(z_t|z_{t-1}, a_{t-1}) p_\xi(r_{t-1}|z_t)$, where $a_t$'s are determined following the behavioral policy $\beta$, distributions $p_\xi(s_t|z_t)$ and $p_\xi(r_{t-1}|z_t)$ follow diagonal Gaussian with mean and covariance determined by MLPs and $p_\xi(z_t|z_{t-1}, a_{t-1})$ follows diagonal Gaussian with mean and covariance determined by LSTM.

Thus, the generative process can be formulated as, i.e., at initialization, $z_0^\xi \sim p(z_0)$, $s_0^\xi \sim p_\xi(s_0|z_0^\xi)$, $a_0 \sim \beta(a_0|s_0^\xi)$; followed by $z_t^\xi \sim p_\xi(\tilde{h}_t^\xi)$, $r_{t-1}^\xi \sim p_\xi(r_{t-1}|z_t^\xi)$, $s_t^\xi \sim p_\xi(s_t|z_t^\xi)$, $a_t \sim \beta(a_t|s_t^\xi)$, with $\tilde{h}_t^\xi = g_\xi[f_\xi(h_{t-1}^\xi, z_{t-1}^\xi, a_{t-1})]$ where $g_\xi$ represents an MLP.

**Training objective.** The training objective for the VAE in MDP is to maximize the evidence lower bound (ELBO), which consists of the log-likelihood of reconstructing the states and rewards, and regularization of the approximated posterior, i.e.,

$$
\begin{aligned}
ELBO(\eta, \xi) = \mathbb{E}_{q_\eta} \Big[ &\sum_{t=0}^T \log p_\xi(s_t|z_t) + \sum_{t=1}^T \log p_\xi(r_{t-1}|z_t) \\
&- KL\big(q_\eta(z_0|s_0)||p(z_0)\big) - \sum_{t=1}^T KL\big(q_\eta(z_t|z_{t-1}, a_{t-1}, s_t)||p_\xi(z_t|z_{t-1}, a_{t-1})\big) \Big].
\end{aligned}
\tag{4}
$$

The proof of Equation 4 is provided in Appendix B.4.

**More discussions on trajectory augmentations.** Latent-model-based models such as VAE have been commonly used for augmentation in offline RL, while they general rarely come with error bounds provided (Hafner et al., 2020; Lee et al., 2020; Rybkin et al., 2021). Prior works have also found that applying generative models to data augmentation can learn more robust predictors that are invariant especially with subgroup identity (Goel et al., 2021). Though generative augmentation models may not perfectly model the subgroup distribution and introduce artifacts, as noted by (Goel et al., 2021), we can directly control the deviations of augmentation from original data with translation or consistency loss as in Equation 4. Our experimental results further show that off-policy selection can benefit more with combination of augmented samples and raw data compared to using raw (original) data only.

## B.4 PROOF OF EQUATION 4

The derivation of the evidence lower bound (ELBO) for the joint log-likelihood distribution can be found below.

*Proof.*

$$\log p_\beta(s_{0:T}, r_{0:T-1}) \tag{5}$$

$$= \log \int_{z_{1:T} \in \mathcal{Z}} p_\beta(s_{0:T}, z_{1:T}, r_{0:T-1}) dz \tag{6}$$

$$= \log \int_{z_{1:T} \in \mathcal{Z}} \frac{p_\beta(s_{0:T}, z_{1:T}, r_{0:T-1})}{q_\alpha(z_{0:T}|s_{0:T}, a_{0:T-1})} q_\alpha(z_{0:T}|s_{0:T}, a_{0:T-1}) dz \tag{7}$$

$$\overset{Jensen's\ inequality}{\geq} \mathbb{E}_{q_\alpha}[\log p(z_0) + \log p_\beta(s_{0:T}, z_{1:T}, r_{0:T-1}|z_0) - \log q_\alpha(z_{0:T}|s_{0:T}, a_{0:T-1})] \tag{8}$$

$$= \mathbb{E}_{q_\alpha}\Big[ \log p(z_0) + \log p_\beta(s_0|z_0) + \sum_{t=0}^{T} \log p_\beta(s_t, z_t, r_{t-1}|z_{t-1}, a_{t-1})$$
$$- \log q_\alpha(z_0|s_0) - \sum_{t=1}^{T} \log q_\alpha(z_t|z_{t-1}, a_{t-1}, s_t)\Big] \tag{9}$$

$$= \mathbb{E}_{q_\alpha}\Big[ \log p(z_0) - \log q_\alpha(z_0|s_0) + \log p_\beta(s_0|z_0) + \sum_{t=1}^{T} \log \big(p_\beta(s_t|z_t)p_\beta(r_{t-1}|z_t)p_\beta(z_t|z_{t-1}, a_{t-1})\big)$$
$$- \sum_{t=1}^{T} \log q_\alpha(z_t|z_{t-1}, a_{t-1}, s_t)\Big] \tag{10}$$

$$= \mathbb{E}_{q_\alpha}\Big[ \sum_{t=0}^{T} \log p_\beta(s_t|z_t) + \sum_{t=1}^{T} \log p_\beta(r_{t-1}|z_t)$$
$$- KL\big(q_\alpha(z_0|s_0)||p(z_0)\big) - \sum_{t=1}^{T} KL\big(q_\alpha(z_t|z_{t-1}, a_{t-1}, s_t)||p_\beta(z_t|z_{t-1}, a_{t-1})\big)\Big]. \tag{11}$$

$$\square$$

## C   MORE EXPERIMENTAL SETUP

### C.1   TRAINING RESOURCES

All experimental workloads are distributed among 4 Nvidia RTX A5000 24GB, 3 Nvidia Quadro RTX 6000 24GB, and 4 NVIDIA TITAN Xp 12GB graphics cards.

### C.2   IMPLEMENTATIONS AND HYPER-PARAMETERS

For FQE, as in (Le et al., 2019), we train a neural network to estimate the values of the target polices $\pi \in \mathbf{\Pi}$ by bootstrapping from the learned Q-function. For DualDICE, we use the open-sourced code in its original paper. For MAGIC, we use the implementation of (Voloshin et al., 2019). For trajectory augmentation, for the components involving LSTMs, which include $q_\alpha(z_t|z_{t-1}, a_{t-1}, s_t)$ and $p_\beta(z_t|z_{t-1}, a_{t-1})$, their architecture include one LSTM layer with 64 nodes, followed by a dense layer with 64 nodes. All other components do not have LSTM layers involved, so they are constituted by a neural network with 2 dense layers, with 128 and 64 nodes respectively. The output layers that determine the mean and diagonal covariance of diagonal Gaussian distributions use linear and softplus activations, respectively. The ones that determine the mean of Bernoulli distributions (*e.g.*, for capturing early termination of episodes) are configured to use sigmoid activations. For training, in subgroups with sample size greater than 200, the maximum number of iteration is set to 1000 and minibatch size set to 64, and 200 and 4 respectively for subgroups with sample size less than or equal to 200. Adam optimizer is used to perform gradient descent. To determine the learning rate, we perform grid search among $\{1e-4, 3e-3, 3e-4, 5e-4, 7e-4\}$. Exponential decay is applied to the learning rate, which decays the learning rate by 0.997 every iteration. For OPE, the model-based methods are evaluated by directly interacting with each target policy for 50 episodes, and the mean of discounted total returns ($\gamma = 0.9$) over all episodes is used as estimated performance for the policy.

### C.3   OPE STANDARD EVALUATION METRICS

*Absolute error* The absolute error is defined as the difference between the actual value and the estimated value of a policy:

$$AE = |V^\pi - \hat{V}^\pi| \tag{12}$$

where $V^\pi$ represents the actual value of the policy $\pi$, and $\hat{V}^\pi$ represents the estimated value of $\pi$.

*Mean absolute error (MAE)* The MAE is defined as the average value of absolute error across $|\mathbf{\Pi}|$ target (evaluation) policies:

$$MAE = \frac{1}{|\mathbf{\Pi}|} \sum_{\pi \in \mathbf{\Pi}} AE(\pi). \tag{13}$$

### C.4   MORE DETAILS ON THE SEPSIS ENVIRONMENT

We also follow (Namkoong et al., 2020) in terms of configuring the reward function and behavioral policy. Specifically, a reward of -1 is received at the end of horizon ($T = 5$) if the patient is deceased (*i.e.*, at least three vitals are out of the normal range), or +1 if discharged (when all vital signs are in the normal range without treatment). The behavioral policy takes the optimal action with 85% of chance, and otherwise switch the vasopressor status.

