# OpenReview forum: "Universal Off-Policy Selection for Human-Centric Systems via Participant Sub-grouping"
_ICLR.cc/2024/Conference — Submitted to ICLR 2024_

### Official Review · Reviewer_6fct · 2023-10-29

**Soundness:** 3 good
**Presentation:** 2 fair
**Contribution:** 2 fair
**Rating:** 5
**Confidence:** 4

**Summary:**

The paper presents an off-policy selection (OPS) method, which aims to determine the best policy from a set of predefined policies. In contrast to the conventional approach of selecting a universal policy, this paper suggests an initial step of clustering trajectories (or, equivalently, participants) and subsequently selecting the most suitable policy for each sub-group. Additionally, the paper introduces a data augmentation technique to address situations where the number of participants within each group is insufficient to accurately estimate policy value. The evaluation is performed in two settings: an offline setting for intelligent tutoring and an off-policy setting for simulated sepsis treatment.

**Strengths:**

- The general idea of not treating everyone with a single policy sounds reasonable and may protect underrepresented groups.
- Evaluations are conducted on both real and simulated data.
- An observation that a fixed policy for everyone may not work well for some sub-groups is interesting.

**Weaknesses:**

- The clarity of the paper could be improved. It was hard for me to follow the details of the paper both the methodology and experiments at some points.
- At the outset, the idea of selecting a policy out of many for each sub-group sounds like designing a new policy. I have difficulty understanding whether, during the partitioning step, any information unavailable to the policy is being used or not. In fact, if there is some information, like patient characteristics, which are used to cluster participants but were not used in training the policy, why not incorporate them in the first place to train the policy?
- There are some inconsistencies in the problem formulation and explanations. Please refer to my questions.
- The choice objective to choose partitioning requires further motivation. Please refer to my questions.
- On the evaluation side, some values need further clarification. For instance, true reward, or AE of OPE.
- The writing could be improved as there are many typos in the text. For example: "it it" in the abstract, "a initial" in Assumption 2, "an classic" on page 6

Recommendations:
- I recommend applying your technique to real offline data of sepsis treatment. For instance, I chose Raghu et al. from your paper and checked its 2023 citations and it seems there are many real evaluations of the policy values including new reliable methods.
- I recommend discussing the fairness aspect of the problem as your approach might be helpful for underrepresented groups.

**Questions:**

1. Please clarify what information may be available at the time of partitioning and if any why they cannot be used during training.
2. In Problem 1, you say the initial state is given but in calculating $V^\pi$ take the expected value wrt $s_0$. What does it mean?
3. I thought Assumptions 1 and 2 imply a one-to-one correspondence between participants and the distribution of initial state. However, a stronger assumption seems to be made on page 3 last paragraph.
4. I'm having a hard time understanding the notation of $V^\pi_{K_m}$ in Definition 1. What distribution $s_0$ is drawn from?
5. Regarding objective (1), isn't the sum over the second term just the value for behavioral policy?
6. Regarding evaluations, please elaborate what are the values reported on the y-axis of Figure 1 in complete detail. Also, what is a true reward mentioned on page 7 for the IE experiment and how AE is defined on page 8?

**Details Of Ethics Concerns:**

Just to make sure human data are in compliance with an IRB.

---

> ### Author Response · Authors · 2023-11-16
> **Author Responses [Part 1/2]**
>
> We thank the reviewer for recognizing our efforts on carrying out and drawing insights from real-world experiments, as well as pointing out one important possible extension of UOPS toward fairness. Hope our detailed responses below would help improve the clarity on the parts that may have led to confusions/mis-understandings, allowing our work to be thoroughly and rigorously evaluated.
>
> Q1: In fact, if there is some information, like patient characteristics, which are used to cluster participants but were not used in training the policy, why not incorporate them in the first place to train the policy.
>
> R1: This is a great question. Unfortunately **we were not able to integrate the sub-group partitioning (or patient characteristic) information into the offline training stage due to the regulatory limitations in HCSs**. Please refer to our response to Txkj's Q2 for more details on this. Moreover, such a practical limitation further highlights the needs of off-policy evaluation and selection prior to online deployment in HCSs, for safety verification as well as improving the efficiency of online testing, which is the motivation of this work. Please also see our response to Q5 below as it is related to this question.
>
> Q2: The writing could be improved as there are many typos in the text. For example: "it it" in the abstract, "a initial" in Assumption 2, "an classic" on page 6.
>
> R2: Thank you for pointing out this. We have fixed these typos in the most recent revision uploaded.
>
> Q3: I recommend applying your technique to real offline data of sepsis treatment.
>
> R3: Thank you for this suggestion. However, as pointed out in [1] that healthcare databases, such as MIMIC, are commonly used for retrospective studies, making it intractable to obtain the actual returns the policies would lead to (as they cannot be deployed). Moreover, as suggested by [2], simulated environments are helpful for "shedding light on certain operating characteristics and best practices". As a result, we chose to rather follow the counterfactual environment introduced in [1], as it would greatly help **isolate the source of improvement regardless of the evaluation approaches used**. Note that this environment is used by many other existing works as cited in the paper, i.e., Hao et al., 2021; Nie et al., 2022; Tang & Wiens, 2021; Lorberbom et al., 2021; Gao et al., 2023; Namkoong et al., 2020 -- which serve as concrete examples to justify the effectiveness of the proposed methods, since ground-truth performance can be accessed (reported in Table 1). Moreover, **we had validated our method over a real-world intelligent education experiment as a result of testing efforts over years, with data collected over 1,000+ students**, besides the sepsis environment.
>
> Q4: I recommend discussing the fairness aspect of the problem as your approach might be helpful for underrepresented groups.
>
> R4: Thank you for this phenomenal suggestion that extends our work's impact significantly. We had added a discussion specific to the potential applications toward promoting fairness in intelligent education (blue texts in Section 3.2 and Appendix A.6).
>
> Q5: Please clarify what information may be available at the time of partitioning and if any why they cannot be used during training.
>
> R5: Besides the regulatory concern following from our response to Q1 above, sub-grouping human participants is considered a non-trivial problem in general. Simply populating the features used for sub-grouping into the input of learning/decision models (but without actually considering the sub-groups as a conditional variable of the models) may fall short in HCSs, as many types of unobserved confounders can affect the outcome; one the other hand, it is challenging to capture these underlying factors explicitly as well [3, 4, 5]. Given both reasons, our work follows the setup that is practically feasible for most HCSs, followed by the approach addressing the problems of sub-grouping (Section 2.2) and policy selection for each sub-group (Section 2.4).
>
> Q6: In Problem 1, you say the initial state is given but in calculating $V^\pi$ take the expected value wrt $s_0$. What does it mean?
>
> R6: Thank you for this question that helped us to improve the clarity of the problem statement. In Problem 1 we meant the initial states are **observable**, which are **variables** sampled from corresponding $\mathcal{S}_0(i')$ -- the term 'known' we originally used may lead to the impression that they can be reduced to constants, which are not our intentions. We have now updated Problem 1 -- please see the texts highlighted in blue there.

---

> ### Author Response · Authors · 2023-11-16
> **Author Responses [Part 2/2]**
>
> Q7: A stronger assumption seems to be made on page 3 last paragraph.
>
> R7: This is due to the setup we considered under HCSs where **one can only sample the initial state $s_0$ once from corresponding $\mathcal{S}_0$** -- the offline data available to use would only contain one trajectory collected from each participant (i.e., repeated sampling of $s_0$'s would be intractable in practice); this fact had been illustrated in detail in the paragraph under assumption 2. Combining it with assumptions 1 and 2, then comes the setup stated on page 3 last paragraph.
>
> Q8: I'm having a hard time understanding the notation of $V_{K_m}^\pi$ in Definition 1. What distribution $s_0$ is drawn from?
>
> R8: Following assumptions 1 and 2, each the $i$-th participant's initial state is drawn from a distribution specific to $i$, i.e., $\mathcal{S_0}(i)$. Then in definition 1 we consider the initial states to be sampled from a union of $\mathcal{S_0}(i)$'s whose corresponding participant belongs to sub-group $K_m$, i.e., $s_0 \sim \bigcup_{i\in\mathcal{I_m}} \mathcal{S_0}(i)$ and the definition of $\mathcal{I}_{m}$ can be found in the main body of definition 1.
>
> Q9: Regarding objective (1), isn't the same over the second term just the value for behavioral policy?
>
> R9: The definition of $V^\beta_{K_m=k\big(s_0^{(i)}\big)}$ still follows from definition 1, but by replacing the target policy $\pi$ with the behavioral policy $\beta$. Conceptually, it's the value of the behavioral policy specific to sub-group $K_m$. Please note that in definition 1 we refer to $\pi$ as 'a policy', not specific to target policies. This is also aligned with our general setup -- the same policy may have different effects toward multiple sub-groups.
>
> Q10: Regarding evaluations, please elaborate what are the values reported on the y-axis of Figure 1 in complete detail. Also, what is a true reward mentioned on page 7 for the IE experiment and how AE is defined on page 8?
>
> R10: Please see below for the responses to each of the questions above.
>
> In Figure 1(a), the values reported on the y-axis are averaging values of normalized learning gains (NLGs) from students who follow the policy selected by each OPS method, representing the returns of selected policy.
>
> In Figure 1(b), the values reported on the y-axis are returns of the selected policy -- blue bars illustrate the estimated values of the policy by each OPS method, and stars (as well as the green line) represent the true averaged normalized learning gain of the students who follow the selected policy.
>
> The true reward is averaged returns over the students who follow the selected policy by an OPS method, where the return is the normalized learning gain (NLG) [6], calculated by students’ pre- and post-exams scores, as defined in Appendix A.1.
>
> The definition of absolute error (AE) used on page 8 can be found in Appendix C.3.
>
> ------------------------------------------------------------------------------------------------------------------------
>
> **We would greatly appreciate it if the reviewer could update the review if satisfied with our responses. We are also more than happy to respond to any other follow-up questions.**
>
> References
>
> [1] Oberst et al. "Counterfactual off-policy evaluation with gumbel-max structural causal models" ICML 2019.
>
> [2] Lu et al. "Is Deep Reinforcement Learning Ready for Practical Applications in Healthcare? A Sensitivity Analysis of Duel-DDQN for Hemodynamic Management in Sepsis Patients." AMIA Annual Symposium. 2020.
>
> [3] Namkoong et al. "Off-policy policy evaluation for sequential decisions under unobserved confounding." NeurIPS 2020.
>
> [4] Nie et al. "Understanding the Impact of Reinforcement Learning Personalization on Subgroups of Students in Math Tutoring." International Conference on Artificial Intelligence in Education 2023.
>
> [5] Ruan et al. "Reinforcement Learning Tutor Better Supported Lower Performers in a Math Task." arXiv:2304.04933 (2023).
>
> [6] Chi et al. "Empirically evaluating the application of reinforcement learning to the induction of effective and adaptive pedagogical strategies." User Modeling and User-Adapted Interaction 21 (2011): 137-180.

---

> > ### Author Response · Authors · 2023-11-19
> > **Mid-point check-in**
> >
> > As we are stepping into the 2nd half of the discussion period, should the reviewer have any follow-ups, we will try out best to address them in time. If satisfied, we would greatly appreciate the reviewer to update the reviews/acknowledge our responses. We sincerely thank the reviewer again for the efforts devoted to the review process, allowing the work to be thoroughly evaluated and discussed.

---

> > > ### Comment · Reviewer_6fct · 2023-11-19
> > >
> > > Dear author,
> > > Please rest assured that I have read your response and will carefully consider it. I need more time to review the details of your paper and your thorough response, which I appreciate. You can expect to hear back from me early next week, before the discussion period ends.

---

> > ### Comment · Reviewer_6fct · 2023-11-20
> >
> > Thank you for your detailed response.
> >
> > R6: This clarified my question. Thank you.
> >
> > R10: Thank you for clarification.
> >
> > R1:
> > > ... we were not able to integrate the sub-group partitioning (or patient characteristic) information into the offline training stage due to the regulatory limitations in HCSs.
> >
> > If this is an offline training, what kind of regulation does not allow for using patient characteristics at the time of training but allows for using them in the selection of the policy? I can understand there might be barriers for interacting with a policy, but shouldn't training offline policies fall under the same category as offline selection of a policy?
> >
> > R5:
> > > Simply populating the features used for sub-grouping into the input of learning/decision models (but without actually considering the sub-groups as a conditional variable of the models) may fall short in HCSs, as many types of unobserved confounders can affect the outcome.
> >
> > It's not straightforward for me to see why this argument is true.
> >
> > R7: I think the language is unnecessarily complex here. Assumption 1 implies every participant has a unique trajectory. Assumption 2 implies the first state of this trajectory follows a participant's unique distribution. I might be missing something but these two assumptions do not imply that by looking at the first state I can identify the participant. This was true if for no two participants, state distributions have shared support. In other words, this was true if clustering based on the first state could be done perfectly.
> >
> > R8: I see your point and I fully understand what this term is representing. But I don't think the notation is appropriate. First of all, what does union of some distributions mean? I guess you mean a uniformly weighted mixture of distributions. Second, let's look at the equation: $s_0$ is drawn from a union of distributions. That sounds right to me. But what is the condition $k(s_0)=K_m$ then? Again, I understand what you mean, but I don't feel equations reflect that.
> >
> > R9: I'm not convinced with your answer. The behavioral policy $\beta$ is fixed and you are summing up all values for all the subgroups. This should simply be the value of the behavioral policy, which is independent of $k(\cdot)$ and can be taken out of the objective.
> >
> > Overall, I thank the authors for their thorough response. The addition of fairness discussion in Appendix A.5 was nice and some of my questions were answered in the response, so I increased my score. However, I still have serious concerns about the presentation, framework, setting, objective, and reading Reviewer's v9Fb comments added to my concern regarding optimization. I also agree with Reviewer Txkj that a significant contribution of your work is the education dataset and other venues might be a better fit than ICLR. Therefore, I'd like to stick to my assessment that this paper is below the acceptance threshold.

---

> ### Author Response · Authors · 2023-11-22
> **Official Comment by Authors**
>
> We thank the reviewer for getting back to us and glad to learn that our initial responses addressed part of the concerns. Although there exist some discrepancies between what the authors introduced in the initial responses and the latest comments from the reviewer above (i.e., Q11 and Q12), **the authors found that all other remaining questions can be addressed through a revision, and hope the responses below and the revision can further clarify them**.
>
> Q11: If this is an offline training, what kind of regulation does not allow for using patient characteristics at the time of training but allows for using them in the selection of the policy?
>
> R11: First, we would like to emphasize that our goal is to use offline data to **evaluate and select policies to be deployed to new participants who join the HCS sequentially, as stated in Problem 1; it addresses a practical issue where the existing line of OPS/OPE research cannot [1-10]**, while offline training is outside the scope of this work. Moreover, as in many experiments with human participants, our IRB requires comprehensive scrutiny over policies' theoretical ground as well as empirical performance before deployment -- **as required by IRB, we were only able to use vanilla DQN to generate candidate target policies, and any changes made to the DQN would be prohibited** (including first sub-grouping the participants and then only training a specific DQN policy for each group).
>
> Q12: It's not straightforward for me to see why the argument in R5 is true.
>
> R12: We were not sure why the reviewer ignored the last part of the sentence following the semicolon in our original R5. References [11-13] illustrated the importance of grouping human participants. Moreover, as stated in Problem 1, our goal is set to be OPS and **offline training with sub-grouping is outside of the scope of this work. However, we believe that this would be a promising avenue for future works to follow on, if not bounded by practical limitations.**
>
> Q13: I might be missing something but these two assumptions do not imply that by looking at the first state I can identify the participant. This was true if for no two participants, state distributions have shared support. In other words, this was true if clustering based on the first state could be done perfectly.
>
> R13: We have verified with the data from the real-world IE experiment that each of the ~1,200 students has a unique initial state, following the state space constituted by 142 expert-extracted features (see Appendix A.1). This implies that the initial state distributions may share very limited support across the cohort (as the population there is considered substantial than typical HCSs), resonating the fact that each student would have a distinct education history, skillset, etc. when starting a course (i.e., each person in the world has unique characteristics and past experiences). We have updated the corresponding texts (in red) there to reflect this.
>
> Q14 (follow-up from original R8): I see your point and I fully understand what this term is representing. But I don't think the notation is appropriate…. what is the condition $k(s_0)=K_m$ then? Again, I understand what you mean, but I don't feel equations reflect tha
>
> R14: We have updated the terms used in definition 1 (in red). Moreover, the definition of $V^\pi_{K_m}$ represents the value of policy $\pi$ **specifically for the sub-group $K_m$**; thus, we used the condition $k(s_0)=K_m$ to illustrate that the expectation is specific to sub-group $K_m$. However, under the updated definition 1 we would not need it anymore.
>
> Q15: I'm not convinced with your answer. The behavioral policy $\beta$ is fixed and you are summing up all values for all the subgroups. This should simply be the value of the behavioral policy, which is independent of $k(\cdot)$ and can be taken out of the objective.
>
> R15: What the reviewer illustrated above follows the typical definition of value functions. In contrast, following from definition 1 and R14 above, in this work we consider the **value functions specific to each sub-group $K_m$**, so we only **take the expectation of the returns with respect to the participants that belong to the same group** $K_m$ in order to define/obtain $V^\beta_{K_m}$.
>
> Q16: reading Reviewer's v9Fb comments added to my concern regarding optimization
>
> R16: In v9Fb's latest response, that reviewer got the idea correctly from our initial responses, and we have updated the corresponding texts (highlighted in red) to clear out the confusion there.

---

> > ### Author Response · Authors · 2023-11-22
> > **Official Comment by Authors**
> >
> > Q17: I also agree with Reviewer Txkj that a significant contribution of your work is the education dataset and other venues might be a better fit than ICLR.
> >
> > R17: We respectfully disagree. There **do exist a number of recent works published in relative venues, which also focus on empirically developing/improving intelligent education systems with OPS and RL techniques** that are published in relative venues, e.g., NeurIPS publications [14-16] from Brunskill's and Finn's groups, three more recent NeurIPS paper [17-19], as well as AAMAS publication [5] from Levine and Brunskill. Consequently, we would tend to believe that ICLR would be a good fit for this empirical/application work, as it also has exceptional records of being inclusive for diversified research topics. **Moreover, we also have a healthcare study in simulation, which has been widely used in related works [20-25].**
> >
> > ------------------------------------------------------------------------------------------------
> >
> > **We greatly appreciate the reviewer's effort and time devoted to the review process, where many meaningful comments have led to improved presentation of our work. Please let us know if our responses above further clarified your remaining concerns. Thank you.**
> >
> > References
> >
> > [1] Fu et al. "Benchmarks for Deep Off-Policy Evaluation." ICLR 2021.
> >
> > [2] Thomas et al. "Data-efficient off-policy policy evaluation for reinforcement learning." ICML  2016.
> >
> > [3] Jiang et al. "Doubly robust off-policy value evaluation for reinforcement learning." ICML 2016.
> >
> > [4] Gao et al. "Variational Latent Branching Model for Off-Policy Evaluation" ICLR 2023.
> >
> > [5] Mandel, Travis, et al. "Offline policy evaluation across representations with applications to educational games." AAMAS. Vol. 1077. 2014.
> >
> > [6] Zhang, Ruiyi et al. "GenDICE: Generalized Offline Estimation of Stationary Values." ICLR 2020.
> >
> > [7] Yang, Mengjiao et al. "Off-policy evaluation via the regularized lagrangian." ICML 2020.
> >
> > [8] Zhang, Shangtong et al. "Gradientdice: Rethinking generalized offline estimation of stationary values." ICLR 2020.
> >
> > [9] Zhang, Michael R. et al. "Autoregressive Dynamics Models for Offline Policy Evaluation and Optimization." ICLR 2021.
> >
> > [10] Nachum, Ofir et al. "Dualdice: Behavior-agnostic estimation of discounted stationary distribution corrections." NeurIPS 2019.
> >
> > [11] Yin, Changchang et al. "Identifying sepsis subphenotypes via time-aware multi-modal auto-encoder." Proceedings of the 26th ACM SIGKDD international conference on knowledge discovery & data mining 2020.
> >
> > [12] Yang, Xi et al. "Student Subtyping via EM-Inverse Reinforcement Learning." International Educational Data Mining Society 2020.
> >
> > [13] Nie, Allen, Ann-Katrin Reuel, and Emma Brunskill. "Understanding the Impact of Reinforcement Learning Personalization on Subgroups of Students in Math Tutoring." International Conference on Artificial Intelligence in Education 2023.
> >
> > [14] Liu, E., Stephan, M., Nie, A., Piech, C., Brunskill, E., & Finn, C. "Giving Feedback on Interactive Student Programs with Meta-Exploration." NeurIPS 2022.
> >
> > [15] Nie, A., Brunskill, E., & Piech, C. "Play to grade: testing coding games as classifying Markov decision process." NeurIPS 2021.
> >
> > [16] Mu, T., Goel, K., & Brunskill, E. "Program2Tutor: combining automatic curriculum generation with multi-armed bandits for intelligent tutoring systems." NeurIPS 2017.
> >
> > [17] Ahmed, U. et al. "Synthesizing tasks for block-based programming." NeurIPS 2020.
> >
> > [18] Piech, C. et al. "Deep knowledge tracing." NeurIPS 2015.
> >
> >
> >
> > [19] Piech, C. et al. "Learning program embeddings to propagate feedback on student code." ICML 2015.
> >
> > [20] Hao, Botao, et al. "Bootstrapping fitted q-evaluation for off-policy inference." ICML 2021.
> >
> > [21] Nie, Allen, et al. "Data-Efficient Pipeline for Offline Reinforcement Learning with Limited Data." NeurIPS 2022.
> >
> > [22] Tang, Shengpu, and Jenna Wiens. "Model selection for offline reinforcement learning: Practical considerations for healthcare settings." MLHC 2021.
> >
> > [23] Lorberbom, Guy, et al. "Learning generalized gumbel-max causal mechanisms." NeurIPS 2021.
> >
> > [24] Gao, Ge, et al. "HOPE: Human-Centric Off-Policy Evaluation for E-Learning and Healthcare." AAMAS 2023.
> >
> > [25] Namkoong, Hongseok, et al. "Off-policy policy evaluation for sequential decisions under unobserved confounding." NeurIPS 2020.

---

### Official Review · Reviewer_v9Fb · 2023-11-01

**Soundness:** 2 fair
**Presentation:** 2 fair
**Contribution:** 3 good
**Rating:** 5
**Confidence:** 2

**Summary:**

The authors propose a policy selection algorithm to produce more optimal behavior policies for human centered systems (HCS). The algorithm uses previously collected offline data with a partitioning function to select optimal policies for each partition of the offline data. New users of the HCS are then assigned to the most similar partition and given the previously selected policy for the partition. The author's primary contribution is their UOPS framework which bridges the gap between online policy deployment and offline policy selection. To support their contribution the authors provide two empirical experiments.

**Strengths:**

* The authors provide two substantial experiments (one real world and one simulated) where their proposed method outperforms 18 alternative methods selected by the authors
* The authors provide extensive connections to existing literature and bring together many ideas from disparate fields such as unsupervised learning (i.e. clustering), off-policy evaluation, and human centered systems.

**Weaknesses:**

* The motivation for the method seems weak. For example, one proposed problem is the time and cost to collect data, however the proposed method still requires trajectories to be collected a priori thus the time and cost of data collection is not removed.
* Sometimes the paper says clustering is done based on the initial state but it does not seem obvious that optimizing (1) requires similarity in the initial state.

**Questions:**

* It is not clear to me how the clustering method suggested optimizes (1). The TICC clustering method maximizes the likelihood that an example belongs to a group correct?
* The terminology of "policy selection" instead of what the more common "policy evaluation" is a little confusing. There does not seem to be any reason why this method couldn't be referred to as an improved policy evaluation technique.
* How does this work relate to Konyushova, Ksenia, et al. "Active offline policy selection." Advances in Neural Information Processing Systems 34 (2021): 24631-24644.
* Why do the authors believe clustering improves performance? If the initial state is unique to participants wouldn't it be possible to learn a single policy that performs well across all states? Why do they think this doesn't happen? Is the policy class being trained on offline data not rich enough?

**Details Of Ethics Concerns:**

No ethics concerns

---

> ### Author Response · Authors · 2023-11-16
> **Author Responses [Part 1/2]**
>
> We are deeply grateful that the reviewer dived into granular details of our work, and brought up many thoughtful questions. Please see our responses below. **We would be more than happy to further respond to any follow-ups the reviewer may have, and would greatly appreciate the reviewer to update the review if our work is worthy for a higher rating.**
>
> Q1: The motivation for the method seems weak -- the proposed method still requires trajectories to be collected a priori thus the time and cost of data collection is not removed.
>
> R1: In both abstract and introduction we emphasized that **online testing** of policies obtained from offline RL training are costly. For example, it would be intractable to deploy all the policies trained with different hyper-parameter sets directly to participants in an intelligent tutoring experiment, since only a limited number of policies can be deployed in a semester. In contrast, **with OPS/OPE one could first estimate how the policy candidates would perform and only deploy the promising ones to online testing**, saving many iterations that would have been used on deploying the policies with sub-par performance. If our writing led to such a confusion, we would be more than happy to further clarify these parts in the manuscript.
>
> Q2: Sometimes the paper says clustering is done based on the initial state but it does not seem obvious that optimizing (1) requires similarity in the initial state.
>
> R2: Thank you for this thoughtful question. We intentionally clustered only over the initial state, because it would be the only information available upon deployment (following Problem 1), i.e., one needs to determine which sub-group each new participant belongs to, at the time when only initial states are observable, in order to deploy the corresponding policy that works best for that group according to historical experience (i.e., the offline dataset). For example, in the intelligent tutoring experiment, the historical data (collected from the first 5 semesters) were used to form clusters over initial states. Then, upon the *beginning* of the 6th semester, only the initial states of the students can be observed, which is the sole information one could use to determine which sub-group within which each student falls.
>
> Q3: It is not clear to me how the clustering method suggested optimizes (1). The TICC clustering method maximizes the likelihood that an example belongs to a group correct?
>
> R3: It takes a two-phase approach to solve (1).
>
> **The 1st phase runs entirely over the offline dataset** -- we run TICC to cluster the initial states (using the data collected from the first 5 semesters) by following its original objective in [1]; it outputs a preliminary partitioning $l: \mathcal{S_0} \rightarrow \mathcal{K}$. Then, the value function difference estimator ${\hat D_{K_m}^{\pi, \beta}}$, from Theorem 1, is trained to estimate $V^\pi_{K_m} - V^\beta_{K_m}$ over all groups $K_m \in \mathcal{K}$ using part of the offline trajectories whose initial state falls in the corresponding group $K_m = l(s_0)$.
>
> **In the 2nd phase, (1) is solved upon deployment** (e.g., at the 6-th semester). The final partitioning $k(\cdot)$ is obtained by exhaustively iterating over all possible cases for each participant $i$ -- plug into $\hat D_{K_m}^{\pi, \beta}$ all policy candidates $\pi \in \Pi$ and all possible partitions $K_m \in \mathcal{K}$ and determine which partition satisfies $\max_{\pi \in \Pi} \hat D_{K_m}^{\pi, \beta}$. Then, assign to participant $i$ the corresponding group, as captured by the mapping function $k: \mathcal{S_0} \rightarrow \mathcal{K}$. We had tried to make this part concise in the manuscript, and can add in more details through a revision, if needed.

---

> ### Author Response · Authors · 2023-11-16
> **Author Responses [Part 2/2]**
>
> Q4: The terminology of "policy selection" instead of what the more common "policy evaluation" is a little confusing.
>
> R4: In lieu of solely estimating (or ranking) the performance of the policy candidates $\pi\in\Pi$ (i.e., what typical OPE does), our work rather considers **how to assign the policy candidates that could potentially work best for each new participant who joins the cohort over time, with only the initial states being observable, as formulated in Problem 1 -- it is a practical problem currently preventing RL from facilitating real-world applications in human-centric systems (HCSs)**; this is illustrated by the intelligent tutoring experiment in Section 3.2. Consequently, we tend to believe that the term "off-policy selection (OPS)" fits better for the topic considered in our work. Moreover, as pointed out in the recent benchmark paper [2], both OPE and OPS are important toward closing the gap between offline RL training and evaluation.
>
> Q5: How does this work relate to Konyushova, Ksenia, et al. "Active offline policy selection." Advances in Neural Information Processing Systems 34 (2021): 24631-24644.
>
> R5: Our response here would be somewhat convoluted with the one above. Konyushova's work followed the typical OPS/OPE setup which targeted to choose the best possible policy using limited historical data, while the same agent is used during collection of historical experience and upon deployment, e.g., the offline dataset is collected from exactly the same type of robot to which the policy chosen by OPS will be deployed. In contrast, our work focuses on resolving the practical issue often encountered in HCSs -- each participant (e.g., student or patient) has a rather distinct background to some extent, so different criteria would be needed to evaluate how well the candidate policies would perform across participants, e.g., some students are visual/auditory learners while others may benefit more from hands-on experiences. This also serves as the motivation for sub-grouping the human participants, as its effectiveness has been demonstrated by other applications in healthcare [3], education [4, 5], etc.
>
> Q6: Why do the authors believe clustering improves performance? If the initial state is unique to participants wouldn't it be possible to learn a single policy that performs well across all states? Why do they think this doesn't happen? Is the policy class being trained on offline data not rich enough?
>
> R6: These are very insightful questions highly related to the philosophy behind the motivation of this work as well as the design of the methodology. Besides the citations above showed that sub-grouping participants would be in general beneficial for HCSs, our experimental analyses (centered around Figure 2) did show that different types of students would benefit from specific types of candidate policies. Moreover, in reality it would be very challenging to collect sufficient data over all different types of participants, due to the nature of HCSs that each one has unique past records/backgrounds. For example, if we denote $\mathcal{S_{K_m}}$ the state space specific to the group $K_m$ (i.e., think of it as a specific type of agent in Mujoco, say HalfCheetah), then the overall state space of the HCS is as large as $(\mathcal{S_{K_m}})^K$, with $K$ being the total number of sub-groups (e.g., the total number of environments Mujoco has). Consequently, the complexity of obtaining a one-size-fit-all policy in HCSs would be analogous to training a single policy that can control any type of robot provided in Mujoco, Adroit, etc. However, we appreciate the reviewer's constructive thoughts to this end and we believe that this would be a very promising avenue for future works to follow up.
>
> [1] Hallac et al. "Toeplitz inverse covariance-based clustering of multivariate time series data" KDD 2017.
>
> [2] Fu et al. "Benchmarks for Deep Off-Policy Evaluation." ICLR 2021.
>
> [3] Baytas et al. "Patient subtyping via time-aware LSTM networks." KDD 2017.
>
> [4] Nie et al. "Understanding the Impact of Reinforcement Learning Personalization on Subgroups of Students in Math Tutoring." International Conference on Artificial Intelligence in Education. 2023.
>
> [5] Yang et al. "Student Subtyping via EM-Inverse Reinforcement Learning." International Educational Data Mining Society (2020).

---

> > ### Author Response · Authors · 2023-11-19
> > **Mid-point check-in**
> >
> > As we are stepping into the 2nd half of the discussion period, should the reviewer have any follow-ups, we will try out best to address them in time. If satisfied, we would greatly appreciate the reviewer to update the reviews/acknowledge our responses. We sincerely thank the reviewer again for the efforts devoted to the review process, allowing the work to be thoroughly evaluated and discussed.

---

> > ### Comment · Reviewer_v9Fb · 2023-11-20
> >
> > I want to thank the authors for their thorough response. There were many papers and ideas I was unaware of (especially [2], very interesting).
> >
> > Unfortunately, after careful consideration I don't see myself changing my score.
> >
> > My primary concern remains the optimization of (1). There is nothing inherently wrong with (1) but it remains unclear to me that it is being optimized even after the further explanation. I don't understand how how TICC optimizes this objective. TICC is an algorithm for time series segmentation and clustering is it not? I'm not an expert on this by any stretch so I could definitely be missing some deeper fundamental connection between these two problems.
> >
> > The method described in the response above makes more sense to me than what I read in the paper. That is, you're optimizing (1) for the deployment partitioning rather than the partition over the training set? And this optimization is done via exhaustive evaluation rather than an algorithm? If that understanding is correct then it wasn't clear in the paper and I'd encourage the authors to consider changing explanation in the section**Optimization over the sub-typing objective** where it is stated that (1) is optimized via TICC.
> >
> > I would like to leave the authors with one final thought. The abstract makes reference to learning style optimization as a key motivator, however, I believe this has largely been shown to be a myth [6]. Perhaps there's a better motivating example that isn't as controversial in the literature?
> >
> > [6] Nancekivell, Shaylene E., Priti Shah, and Susan A. Gelman. "Maybe they’re born with it, or maybe it’s experience: Toward a deeper understanding of the learning style myth." Journal of Educational Psychology 112.2 (2020): 221.

---

> ### Comment · Area_Chair_jovH · 2023-11-20
>
> Dera v9Fb,
>
> The author reviewer discussion period is ending soon this Wed. Does the author response clear your concerns w.r.t., e.g., lack of strong motivation and lack of comparison with related works, or there are still outstanding items that require more discussion?
>
> Thanks again for your service to the community.
>
> Best,
> AC

---

> ### Author Response · Authors · 2023-11-21
>
> We thank the reviewer for getting back to us in time, and carefully considered our prior responses.
>
> *To address your remaining concern on solving (1)* -- you provided the correct illustration of how it is solved above, so we went ahead and update the corresponding paragraph with such details, which is highlighted in red in the latest revision.
>
> *In regards to your other question on the motivator* -- we have now updated this motivator to be aligned with the finding from the real-world IE experiment (centered around Figure 2). The changes are also highlighted in red.
>
> We greatly appreciate reviewer's attention paid to such details, where we believe the comments lead to significantly improved presentation of the paper overall. Moreover, we will strive our best to help clarify any remaining concerns from the reviewer, and make corresponding revisions, if any -- we would be truly grateful for having the chance revising the paper toward the reviewer's standard of satisfaction.

---

> > ### Author Response · Authors · 2023-11-22
> > **Did we address the remaining 2 concerns?**
> >
> > Dear reviewer v9Fb,
> >
> > As the author-reviewer discussion window is closing in a day, we wanted to reach out and learn if our latest responses and the updated manuscript helped address your remaining concerns. Note that today we further included in Appendix B.1 the full details in regards to optimization over (1), so that we are not bounded by the page limitation anymore. Thank you.
> >
> > Sincerely,
> > Authors

---

### Official Review · Reviewer_Txkj · 2023-11-06

**Soundness:** 3 good
**Presentation:** 3 good
**Contribution:** 2 fair
**Rating:** 5
**Confidence:** 3

**Summary:**

The authors propose a framework for accounting for heterogeneity in people when evaluating and selecting RL policies offline. They call their method universal off-policy selection (UOPS). Broadly, the method consists of categorizing each human participant into a certain class, and then finding a policy that suits each participant class the best. In this way, the RL method accounts for heterogeneity among multiple types of participants. The authors demonstrate the empirical performance of their approach on an educational task and a healthcare task.

**Strengths:**

- The authors clearly communicate their objective and the backbone of their approach. They are well-motivated, especially with the education application.
- The educational dataset is novel and is impressive in its breadth.
- The proposed methodology behind UOPS is quite straightforward and intuitive. No extra frills added where not needed, which I appreciate.

**Weaknesses:**

- While the paper is quite convincing in its results on the education dataset, I'm not sure that ICLR is the best venue for these results. The methodology presented is less novel/interesting than the education dataset and results. This leads me to think that this work could be better suited for an education-based venue.
- Consider a simple approach to the same problem: Cluster the students using some basic technique, then run any out-of-the-box RL method on each group independently. How would this compare to your results? It seems that there is a decently large sample size and not a crazy high number of subgroups.

**Questions:**

See the second bullet in the Weaknesses section.

---

> ### Author Response · Authors · 2023-11-16
> **Author Responses**
>
> We are grateful for the reviewer's recognition toward the novelty of our methodology as well as the impact of our real-world experiment. Please see below our detailed responses to the comments. **We would greatly appreciate the reviewer updating the review if satisfied, and we are more than happy to respond to any follow-ups the reviewer may have.**
>
> Q1: While the paper is quite convincing in its results on the education dataset, I'm not sure that ICLR is the best venue for these results.
>
> R1: Thank you for recognizing the impact of the real-world education experiment as a result of years of deployment, data collection, and analyses efforts. We would emphasize that the general goal of OPS/OPE is to bridge the gap between offline RL policy optimization and online testing in many **real-world** domains (as pointed out in [1, 2, 3, 4] etc.), due to safety and cost concerns, e.g., the IRB committee may require the policies to demonstrate sufficient performance before they can be deployed to students/patients. There **do exist a number of recent works published in relative venues, which also focus on empirically developing/improving intelligent education systems with OPS and RL techniques** that are published in relative venues, e.g., NeurIPS publications [5, 6, 7] from Brunskill's and Finn's groups, another recent NeurIPS paper [8], as well as AAMAS publication [9] from Levine and Brunskill. Consequently, we would tend to believe that ICLR would be a good fit for this empirical/application work, as it also has exceptional records of being inclusive for diversified research topics.
>
> Q2: Consider a simple approach to the same problem: Cluster the students using some basic technique, then run any out-of-the-box RL method on each group independently.
>
> R2: Thank you for sharing this very interesting idea that points out one of the future avenues for our work. Please note that our IRB, as in many experiments with human participants, requires comprehensive scrutiny over policies' theoretical ground as well as empirical performance before deployment, so we only have very limited options toward what types of policies can be deployed -- as a result, we were only able to use DQN due to it's straightforward methodology, and there exist many works that have shown it's efficacy in human-centered experiments [10,11,12] etc. This also serves as the main obstacle following your idea above, as directly deploying custom RL policies to students would be intractable. Moreover, please note that sub-grouping human participants is considered a non-trivial problem, with a number of works concurrently exploring this topic concurrently through different approaches over various applications [13, 14, 15] etc., and a main part of our work (Section 2.2) attempted to address this problem in the context of intelligent education. Again, we appreciate the reviewer's thoughtful comment and hope this idea can be further explored in the future.
>
>
> [1] Fu et al. "Benchmarks for Deep Off-Policy Evaluation." ICLR 2021.
>
> [2] Thomas et al. "Data-efficient off-policy policy evaluation for reinforcement learning." ICML  2016.
>
> [3] Jiang et al. "Doubly robust off-policy value evaluation for reinforcement learning." ICML 2016.
>
> [4] Gao et al. "Variational Latent Branching Model for Off-Policy Evaluation" ICLR 2023.
>
> [5] Liu et al. "Giving Feedback on Interactive Student Programs with Meta-Exploration." NeurIPS 2022.
>
> [6] Nie et al. "Play to grade: testing coding games as classifying Markov decision process." NeurIPS 2021.
>
> [7] Mu et al. "Program2Tutor: combining automatic curriculum generation with multi-armed bandits for intelligent tutoring systems." NeurIPS 2017.
>
> [8] Ahmed et al. "Synthesizing tasks for block-based programming." NeurIPS 2020.
>
> [9] Mandel et al. "Offline policy evaluation across representations with applications to educational games." AAMAS 2014.
>
> [10] Raghu et al. "Deep Reinforcement Learning for Sepsis Treatment." NeurIPS 2017.
>
> [11] Liu et al. "Learning the dynamic treatment regimes from medical registry data through deep Q-network." Scientific reports 9.1 (2019): 1495.
>
> [12] Raghu et al. "Continuous state-space models for optimal sepsis treatment: a deep reinforcement learning approach." MLHC 2017.
>
> [13] Namkoong et al. "Off-policy policy evaluation for sequential decisions under unobserved confounding." NeurIPS 2020.
>
> [14] Hamid, Jemila S., et al. "Cluster analysis for identifying sub-groups and selecting potential discriminatory variables in human encephalitis." BMC infectious diseases 10.1 (2010): 1-11.
>
> [15] Liu, An-An, et al. "Hierarchical clustering multi-task learning for joint human action grouping and recognition." IEEE transactions on pattern analysis and machine intelligence 39.1 (2016): 102-114.

---

> > ### Author Response · Authors · 2023-11-19
> > **Mid-point check-in**
> >
> > As we are stepping into the 2nd half of the discussion period, should the reviewer have any follow-ups, we will try out best to address them in time. If satisfied, we would greatly appreciate the reviewer to update the reviews/acknowledge our responses. We sincerely thank the reviewer again for the efforts devoted to the review process, allowing the work to be thoroughly evaluated and discussed.

---

> ### Comment · Area_Chair_jovH · 2023-11-20
>
> Dear Txkj,
>
> The author reviewer discussion period is ending soon this Wed. Does the author response clear your concerns w.r.t., e.g., fit to ICLR and simple baselines, or there are still other outstanding items that you would like to discuss more?
>
> Best,
> AC

---

### Meta-Review · Area_Chair_jovH · 2023-12-05

**Metareview:**

This work proposes to divide offline dataset into subgroups based on certain metrics and then perform group-wise off-policy selection. When new data arrives, a group is first assigned to the data and the best policy in that group is assigned to the new data.

Strengths: The work conducts real-world experiments to demonstrate the efficacy of the proposed method in education.

Weakness: My major concern is the contribution of this work to the general ML community. Indeed, the success in real world education domain is impressive. However, from an ML perspective, the proposed algorithm is not well understood. As suggested by Txkj, a simple approach could be to divide the dataset first and apply any existing OPE RL algorithm group-wise. I understand this might not be possible in the real world due to IRB but there are plenty of generic simulated datasets for generic OPE RL algorithms. Conducting such an ablation study in generic offline dataset is necessary to validate the efficacy of the proposed estimator.

**Justification For Why Not Higher Score:**

An important ablation study is lacking such that the contribution of the work to the general ML community is not clear.

**Justification For Why Not Lower Score:**

N/A

---

### Decision · Program_Chairs · 2024-01-16

Reject